# Groundwater Salinization and Freshening Processes in the Luy River Coastal Aquifer, Vietnam

**Linh Pham Dieu** [1,2,*], **Diep Cong-Thi** [1,2], **Tom Segers** [1], **Huu Hieu Ho** [2], **Frédéric Nguyen** [3,4] **and Thomas Hermans** [1]

1   Laboratory of Applied Geology and Hydrogeology, Department of Geology, Ghent University, Krijgslaan 281-S8, 9000 Ghent, Belgium; diep.congthi@ugent.be (D.C.-T.); tom.segers@ugent.be (T.S.); thomas.hermans@ugent.be (T.H.)
2   Vietnam Institute of Geosciences and Mineral Resources (VIGMR), Hanoi 100000, Vietnam; hohuuhieu@yahoo.com
3   Urban and Environmental Engineering Research Unit, Liege University, Allée de la Découverte 9, 4000 Liège, Belgium; f.nguyen@uliege.be
4   Department of Earth and Environmental Sciences-Geology, KU Leuven, Celestijnenlaan 200e-bus 2411, 3001 Leuven, Belgium
*   Correspondence: linh.phamdieu@ugent.be

**Abstract:** The Vietnamese Binh Thuan province located along the Southern Central part of the coast is one of the driest in the country. The population is relying largely on groundwater for irrigation, and the groundwater resources are threatened by climate change and saltwater intrusion in river estuaries. Recent studies, however, reveal that the extension of saltwater intrusions was larger than expected, raising some doubts on the actual origin of the intrusions. In this study, we use the geochemical characterization of groundwater samples collected in both the dry and rainy seasons in the Luy river coastal area to identify hydrochemical processes responsible for the salinization and variations taking place in the shallow aquifers. The distribution of chemical components of groundwater and the indicators presenting the freshening and salinization processes, such as cation exchange code and chloride conservative element, the ionic delta, the HFE-diagram, and geochemical modelling, were studied. The results show that 65% of the samples exceed the WHO limit for drinking water and 100% of them have a degree of restriction on use from slight–moderate to severe in potential irrigation problems according to FAO. In contrast to previous expectations, freshening is the dominant process in the aquifers, and it is more advanced in the rainy season. Due to a lack of recharge and aquifer exploitation, salinization is more severe in the deeper aquifer and during the dry season. Saltwater intrusion is not limited to the zone close to the river but extends further inland and at depth, and salinity can vary quickly over short distances. Based on these new insights, we develop a new conceptual model for the evolution of salinization in the Luy river catchment involving a natural freshening process of connate water combined with anthropic influence. The conceptual model will form the basis for the development of a groundwater model of the study area and eventually lead to sustainable management scenarios for this coastal region, preventing further deterioration of the groundwater resources.

**Keywords:** hydrochemistry; freshening; saltwater intrusion; coastal aquifer





## 1. Introduction

Saltwater intrusion (SI) is one of the most widespread threats occurring in coastal areas worldwide [1]. It does not only decrease water quality by raising salinity to levels exceeding acceptable drinking and irrigation water standards but also threatens the prospective exploitation of coastal aquifers and soil salinization. A large number of studies have been carried out in coastal areas across the globe to understand the problem of saltwater intrusion, for example, in Africa [2,3], America [4,5], Asia [6,7], Australia [8,9] and Europe [10,11].

Seawater encroachment inland is the most common cause for the increase in salinity. This often occurs in arid and semi-arid regions where groundwater constitutes the main freshwater resource and additional pressures related to climate change, such as sea level rise, changes in recharge patterns and groundwater withdrawal have impacted the hydrologic balance in the coastal aquifers [12–14]. Salinization can also be the result of upconing, where saltwater is present at depth and rises up due to groundwater extraction [15]. The presence of fossil seawater might also be a source of salinity [1,16,17]. This old seawater can primarily be found in low permeability formations, such as clay and silty clay where it stays entrapped in the sediments.

As opposed to saltwater intrusion, freshening processes are also common in coastal aquifers, for example through infiltration of rainwater, reduction in exploitation flowrate or artificial recharge [18–25]. The freshwater recharge gradually infiltrates through the shallow soil toward the aquifers leading to salinity and groundwater chemical changes.

Through its 3444 km long coast, Vietnam is particularly vulnerable to SI. Several local hydrogeochemical studies on saltwater intrusion have been performed in Vietnam [26–31]. Those studies mostly focused on the Red River delta and Mekong delta—the areas located in the north and south of Vietnam, respectively.

Up to now, the southern central coast of Vietnam has received little attention. Investigations by the National Center for Water Resources (NAWAPI) revealed saltwater contamination in both river estuaries and further inland, threatening the availability of freshwater for irrigation and drinking purposes [32,33]. These early investigations delineated the SI zone based on the TDS, and concluded, based on the spatial patterns, that SI was caused by direct infiltration from surface water in the river estuaries. However, these conclusions were derived from mostly shallow boreholes, without a detailed analysis of the geochemical composition. A recent geophysical study in the Luy River catchment based on electrical resistivity tomography (ERT) [34] revealed the deeper part of the coastal aquifer along the downstream part of the Luy River is mostly occupied by saltwater, whereas the distribution of salinity in the shallow part is extremely variable. Large variations of salinity are observed at short distances, without any apparent correlation with the distance to the Luy river and its tributaries, highlighting a complex distribution of salinity. This geophysical survey showed that the estimation from NAWAPI for the extension of the SI zone was underestimated and that the current situation was likely not the result of recent river water infiltration only, but a consequence of the evolution of the context in the catchment involving both natural and anthropic processes. The main objective of this article is to identify the origin of saltwater intrusion in the Luy River catchment, as this constitutes a first necessary step to propose a sustainable management of the groundwater resources in the study area. Our recent geophysical data raised some serious doubts about the origin of the salinity (river infiltration), but it is not sufficient on its own to draw definitive conclusions. Therefore, we propose a thorough hydrochemical analysis of newly acquired water samples in order to identify which processes are taking place and what is the origin of the high salinity. This will eventually allow us to propose a new conceptual model of the aquifer in agreement with both geophysical and geophysical data.

In coastal aquifers, the situation is often very complex with several saltwater intrusion and freshening processes playing a role at the same time [1]. Although the mapping of the total dissolved solid content (TDS), as can be obtained from geophysical data [34–37], is a first necessary step, it is often not sufficient to reveal the complexity and the dynamics of the processes taking place [38]. Identifying factors governing groundwater geochemistry and the identification of the stage and cause of salinization will assist in efficiently evaluating the suitable groundwater quality for irrigation and drinking water purposes and to propose adapted management strategies [39]. The most direct way to investigate contamination by seawater intrusion and freshening processes is to use indicators derived from groundwater chemical components [38–40]. The indicators of cation exchange processes, such as cation exchange codes [21,41] and chloride (Cl) conservative element [39], are of particular interests as they can reveal if freshening or salinization processes are dominant

and even the current stage of the process [19]. Similarly, the evolution of the hydrochemical facies can evaluate how saltwater intrusion is evolving with time and space [42], although there might be a delay between a change in the drivers of saline intrusion and its effect on the hydrochemical signature, the response to salinization being faster than the one to freshening [21].

For the objective of this study, we perform a new hydrogeochemical survey in the Luy river catchment, including both shallow and deep wells and samples collected during both the dry and wet season. Our study corresponds to the first in-depth hydrochemical analysis of the seawater freshening/intrusion processes based on water composition as well as the first attempt to identify their seasonal variations. In addition, we propose an integrated interpretation of the new datasets with the available geophysical results to distinguish the effect of recent saltwater intrusion from freshening processes that are still the dominant processes controlling water quality at the scale of the catchment. The final result of our study is a conceptual model of seawater intrusion, taking into account natural and anthropic effects with the effect of seasonal cycles, which serves as a basis for future groundwater modelling and management.

## 2. Study Area Description

### 2.1. Topography, Land Use and Surface Water

The study area is the alluvial plain of the Luy River located in the two coastal districts of Tuy Phong and Bac Binh, Binh Thuan province (Figure 1). The population in 2018 of these districts is 269,249 inhabitants with a density of 102 inhabitants/km$^2$ [43]. The study area belongs to the Luy river basin, which originates from high mountains in the eastern border of the Di Linh Plateau in the Lam Dong province and flows to the Phan Ri river mouth. In 2008 and 2009, the Dai Ninh (Lam Dong province) and Bac Binh (Binh Thuan) hydropower plants were constructed in the upstream part of the Luy river. The hydropower reservoirs are recharged by water from the Dong Nai river system and these reservoirs play a significant role in the management of the freshwater supply to the Luy river basin during the dry season, providing irrigation water for the production of rice in the study area. However, the dam overflow resulting from heavy rains during the rainy season combined with water releases from these hydropower stations also sometimes cause floods in the study area [44]. In addition, riverine SI occurs in the downstream part through the Luy river estuary during high tides. TDS measurements in the Phan Ri river mouth during the summer 2019 show values of 15 to 20 g/L up to 10 km inland (Figure 1). These measurements were recorded during high tide when the TDS value is supposed to be the most affected by SI from the Luy river estuary.

The study area is bounded in the northwest by a mountainous area with altitude ranging from 1200 to 2000 m above sea level which is covered by 70 to 100% of forest. The elevation declines sharply from 1200 to 300 m and then regularly from 300 to 100 m into the coastal plain. The south of the study area is delineated by the red sand dune, adjacent to the Vietnamese Eastern Sea, with elevation ranging from 100 to 300 m. The southeast of the study area is limited by the white sand dune complex, also adjacent to the Vietnamese Eastern Sea. The alluvial plain is relatively flat, with the elevation sloping gently from west to east. The main crops cultivated in this plain are dragon fruit plants and rice depending on the water supply and irrigation network in each location. Rice is typically cultivated where irrigation water from dams is available, whereas dragon fruit, which is more drought and salinity resistant, is cultivated in other areas, generally relying on groundwater for irrigation.

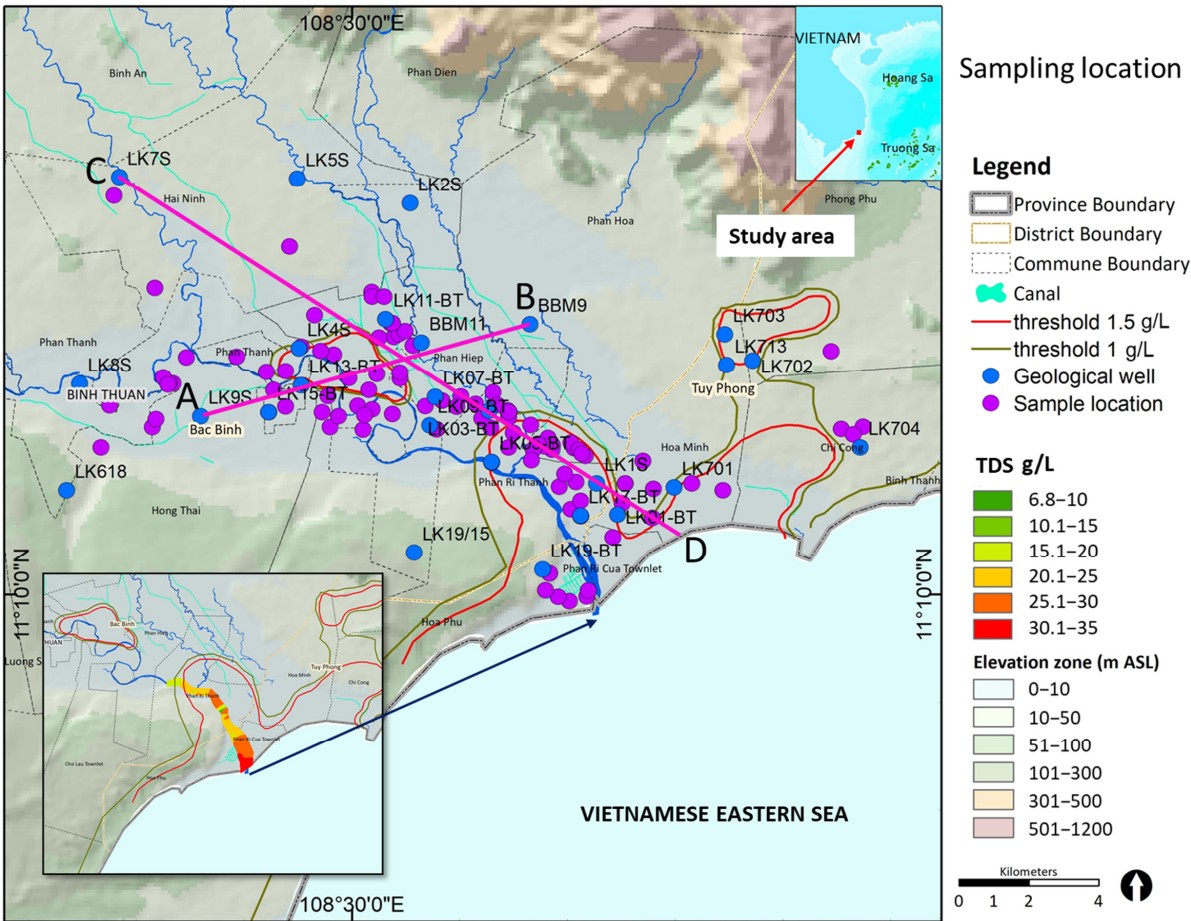

**Figure 1.** The location of the study area with the topography, sample location and TDS measurement in the Phan Ri river mouth in the summer of 2019; AB and CD solid cross section lines.

*2.2. Climate*

The Binh Thuan province has a typical semi-arid climate with the rain season from May to October and the dry season from November to April. The climate is characterized by low rainfall, high evaporation and strong land winds. The 1982–2019 meteorological data collected in the Phan Thiet station located 50 km south of the study area shows that the mean monthly temperature varies from 25.2 °C in January to 28.7 °C in May. The average monthly humidity changes from 75.1% in January to 84% in September. The average monthly evaporation varies from 91.2 mm in October to 140.8 mm in March whereas the average monthly rainfall reaches its highest value of 194 mm in September and minimum value of 0.4 mm in February.

The Tuy Phong and Bac Binh districts have 6 dry months per year with less than 100 mm of monthly rainfall. With a variety of typical deserted lands including sand, stone or salt deserts and degraded land, the province is facing desertification, which has a strong impact on overall agricultural production, the environment and socio-economic activities [45–47].

*2.3. Geology and Hydrogeology*

The Luy river plain is mostly covered by fluvial–marine sediments originating from the Pleistocene and Holocene (Figure 2). The fluvial–marine sediments from the middle to late Pleistocene are distributed in the higher parts of the plain characterized by sequences of round gravel, sand and grits or sand intercalating with silt and clay forming terraces of the Luy river tributaries. The middle–late Holocene sediments are found in the lower part of the plain which is mainly made up of clay, clayish silt mixed sand, brown yellowish or

greyish silty sand containing various amounts of foraminifera, shell fragments, lithoclasts and plant fragments [48].

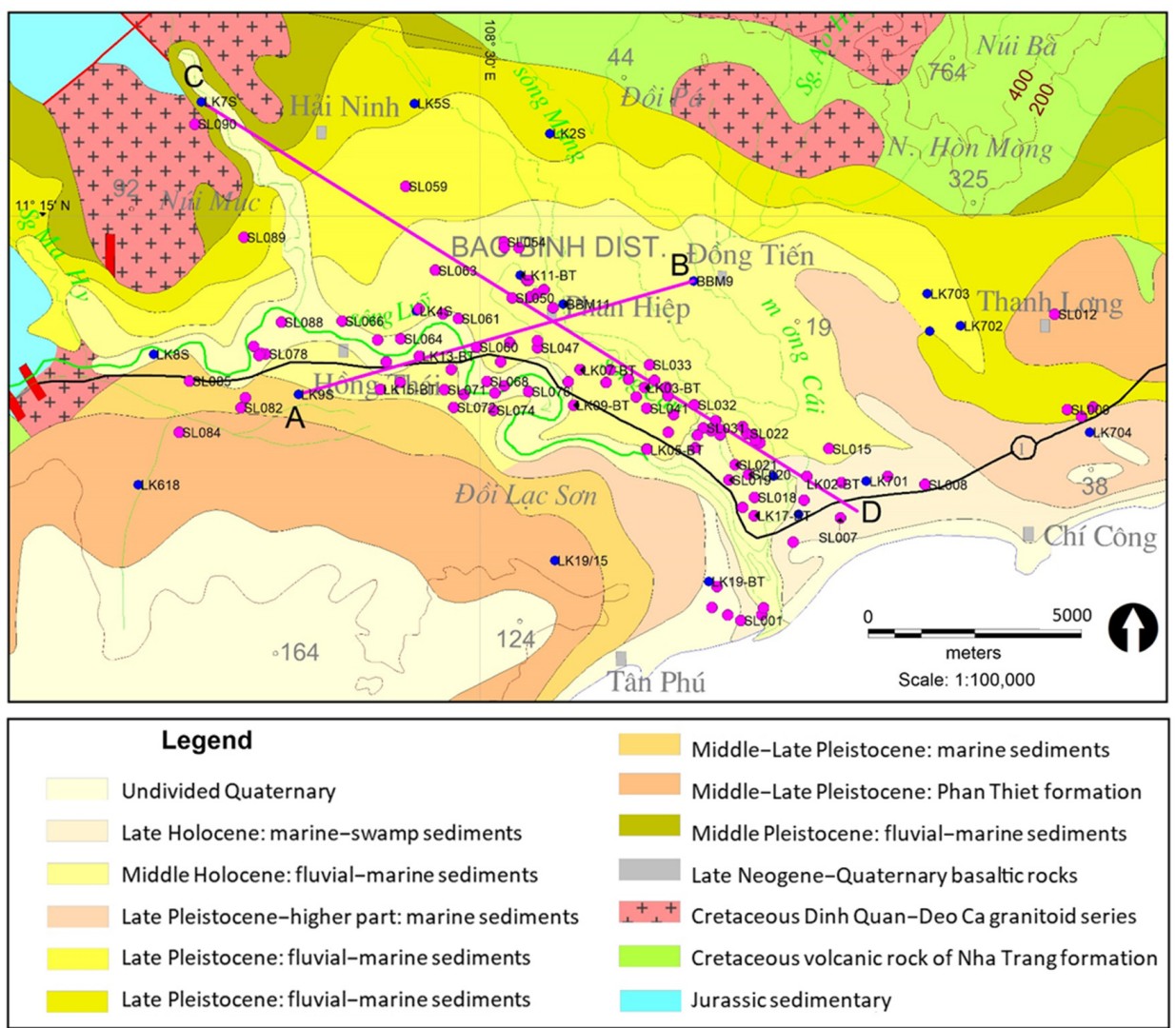

**Figure 2.** Geological map of the study area (modified from Hoang P., 1997), sample locations are the purple dots whereas blue dots are well log locations.

The study area underwent cyclic transgressive and regressive events from the Pleistocene to the Holocene [49]. Of particular importance is the sea-level high-stand that started between 6721 and 5869 calibrated years before present and the warming of coastal waters during the mid-Holocene [50,51]. These events might have influenced the sedimentary composition and caused fossil seawater to be trapped in the heterogeneous sediments in the Luy river delta.

This study focuses on the porous aquifers, including the unconsolidated sediments of both the Holocene and Pleistocene. The thickness of the Holocene aquifer is thin in the right bank (2–4 m) but it increases in the left bank from 2 to 20 m. In contrast, the Pleistocene aquifer is thick on the right bank, due to the high altitude of the sand dune, and is thinner on the left bank.

The Holocene and Pleistocene aquifers can sometimes be distinguished as an unconfined aquifer and a confined aquifer, respectively, when semi-continuous clay layers and lenses are present, but otherwise they just lie on each other sequentially (Figure 3b,c). It is underlined by granite and granitoid from the Cretaceous and sandstone, claystone and

siltstone from the Jurassic. These layers have a limited water storage capacity and are considered as secondary aquifers in the study area.

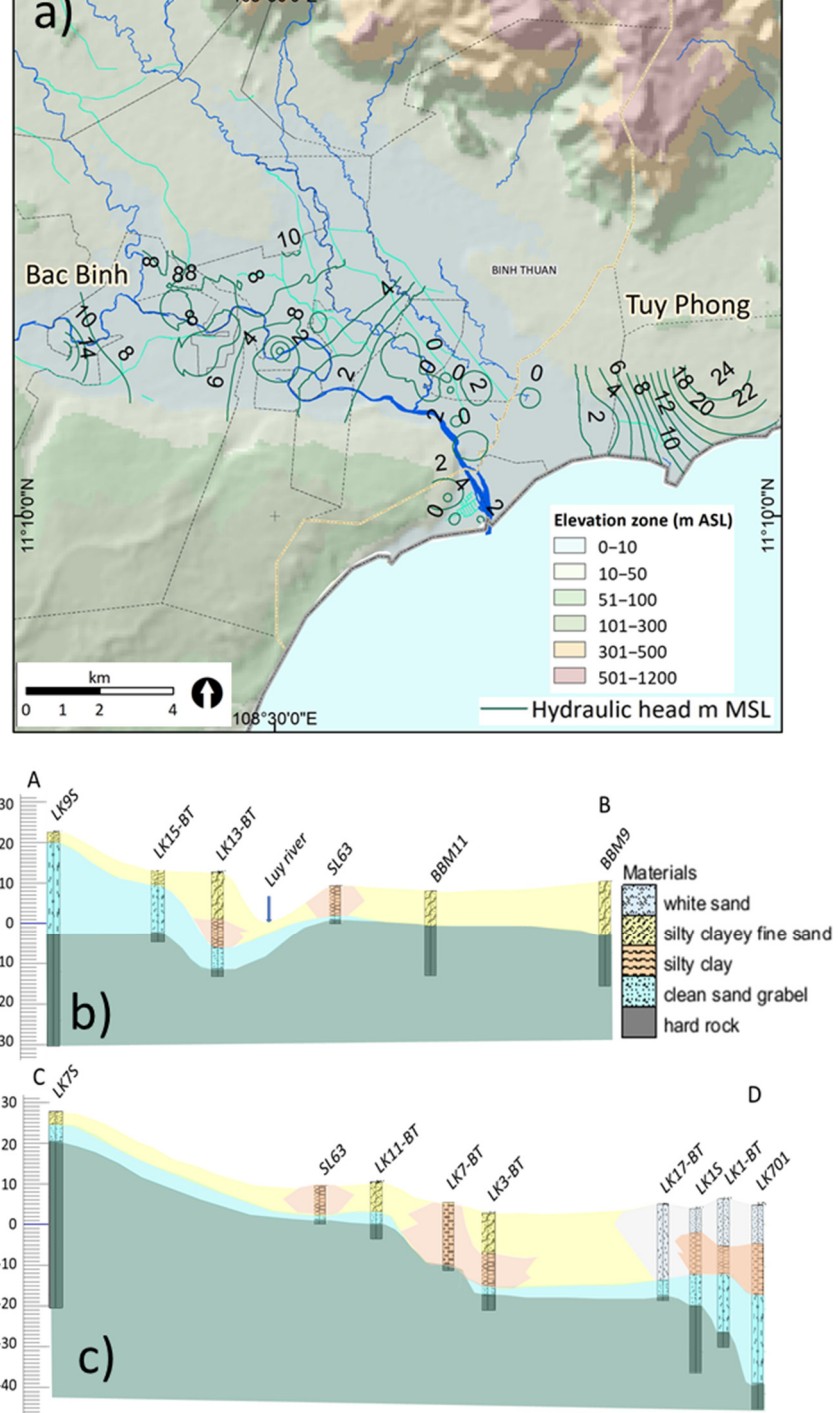

**Figure 3.** (**a**) The elevation of the water table in the study area measured in the summer 2020 and the hydrostatigraphic cross sections along line AB (**b**) and CD (**c**). The white and red sand dune complexes are characterized by higher water levels. In the upstream part, the piezometric table remains above sea level whereas in the central of the plain and near the coast, the water table is close to sea level or sometimes even below.

Water recharge to the aquifers mainly comes from precipitation during the wet season and surface water including through irrigation practices. Outflows occur at the Luy

river estuary and underground discharge to the sea, whereas abstractions occur through pumping for irrigation, aquaculture farming and mining activities. The construction of the hydropower reservoirs in the upstream part since 2008 has been increasing the freshwater supply to the aquifer as it also supplies the study area with fresh surface water through irrigation channels. On the contrary, it might have increased riverine SI by reducing the average flowrate in the Luy river.

Figure 3a illustrates the piezometric map in the study area expressed in freshwater head above the reference level (mean sea level at elevation 0). The main hydraulic gradient is along the Luy river with a secondary component perpendicular to it. Note that locally the natural gradient is perturbed due to water abstractions or irrigation practices.

## 3. Methods

Our methodology is based on newly acquired water samples all over the alluvial plain of the Luy River catchment. The extension of the study area was based on previous data that were available (previous samples and geophysical data) to cover the zones where saltwater intrusion is encountered. The upstream part of the Luy river basin is known to be fresh [32] and SI is known to be the main issue for groundwater quality, whereas the presence of trace elements such As is not expected. The analysis of the hydrochemistry of the water sample is used to identify the origin of saltwater intrusions. In Section 5, the results of the hydrochemical analysis will be combined with existing data to derive a conceptual model of the SI processes in the study area.

### 3.1. Water Sampling

The objective of the sampling campaign was to provide a better coverage of the unconsolidated aquifer than previous studies [32,33]. Ninety groundwater samples were collected in the summer of 2020 from both dug wells and drilled wells owned by farmers using the USGS protocol [52]. These wells are typically used for irrigation purposes and are often abandoned by the farmers when they become too saline. They are all potentially factors of SI by inducing lateral flows or upconing if saline water is present in the vicinity. Because those samples were mostly shallow (<5 m: 6 samples—7%, from 5 to <15 m: 69 samples—76%, >15 m: 9 samples—10% and 6 samples—7%: no information of depth) and that ERT data [34] identified a clear distinction between the shallow and deeper aquifer, 10 nested wells with screen intervals in both the shallow and deep parts of the aquifer were drilled. These are forming the new monitoring network in the Luy river catchment and provide 20 additional samples. In addition, most wells were revisited in April 2021 during the dry season. Wells in which the TDS (converted from measured electrical conductivity—EC in the field with TDS (mg/L) = 0.65 × EC (μS/cm)) was significantly different with an absolute difference greater than 500 mg/L (corresponding to 15 wells) and the wells of the monitoring network were resampled for comparison with the wet season. The location of the samples is shown in Figure 1. For major cation and anion analysis, the samples were collected through a 0.45 μm cellulose acetate filter. Samples for cation analysis (Al, Ca, Fe, K, Mg and Na) were filtered and collected into 50 mL polyethylene (PE) bottles and acidified to 1% with $HNO_3$. Samples for anion analysis (F, Cl, Br, $SO_4$) were filtered and collected into 50 mL PE bottles with no further treatment. Samples for $NO_3$, $NO_2$ and $PO_4$ determinations were filtered and collected into 50 mL polyethylene (PE) bottles adding 1% chloroform. Samples for determining the total dissolved inorganic carbon (DIC) were not filtered but were collected into 100 mL bottles and were analyzed on the same day.

All bottles were rinsed 3 times before sampling and completely filled with the sample to avoid air contact and afterward stored at a temperature of +5 °C and sent to Belgium for analysis in the Laboratory for Applied Geology and Hydrogeology, Department of Geology, Ghent University.

### 3.2. Chemical Analysis

Major cations ($Na^+$, $K^+$, $Ca^{2+}$, $Mg^{2+}$, $Fe_{Total}$ and $Mn_{Total}$) were analyzed by inductively coupled optimal emission spectrometry (ICP-OES) and major anions ($Cl^-$, $SO_4^{2-}$, $NO_3^-$, $PO_4^{3-}$, $NH_4^-$ and $NO_2^-$) were analyzed using a spectrophotometer (Shimadzu UV1800). $Br^-$ and $F^-$ anions were analyzed by an ion-specific electrode. pH was measured on-site using a pH electrode calibrated against commercial buffer solutions. The dissolved inorganic carbon (DIC) concentrations were measured using an alkalinity titration.

Among 110 water samples, 54 samples have an ionic balance error within $\pm 5\%$ whereas the charge balance of 43 samples range within $\pm 5$–$10\%$. Twelve samples have an ionic balance error between $\pm 10$–$20\%$ and one sample has an ionic balance error of 41%. The latter was completely left out of the analysis process. In this study, samples with an ionic balance error up to 20% were kept for analysis to maximize the number of samples. The high error for some samples is attributed to the difficult sampling conditions, the conservation of the samples during transport and the dilution of the samples for analysis because of the high salinity. The ionic balance for the 33 samples collected during the dry season is below 10%.

### 3.3. Geochemical Analysis

The analysis of the data involved four steps: (1) the classification of water using the Stuyfzand's classification [41]; (2) the evolution of hydrochemical facies using [53] and HFE-D diagrams [54]; (3) the estimation of the origin of solutes using conservative element behavior and calculation of aqueous speciation and mineral saturation indices; (4) the calculation of the expected composition based on conservative mixing.

#### 3.3.1. Stuyfzand's Classification

In Stuyfzand's classification, the primary type of water uses the $Cl^-$ content (in mg/L) corresponding with the codes <150 = F-Fresh, 150–300 = Fb-Fresh-brackish, 300–1000 = B-Brackish, 1000–10,000 = Bs-Brackish-salt, 10,000–20,000 = S-Salt and >20,000 = H-Hyperhaline. We also use Stuyfzand's cation exchange code which determines if cation exchange has taken place and the original component of the exchange. According to Stuyfzand [41], the surplus of (Na + K + Mg) often points at a (former) freshwater encroachment and the deficit of (Na + K + Mg) often points at a (former) saltwater intrusion. The cation exchange code also plays a significant role in the examination of salinization and freshening processes because it characterizes the composition of end-members. In the coastal aquifers, seawater (SW) and freshwater (FW) are considered as two end-members that are characterized by the water types of S-NaCl0 and F-CaHCO$_3$0, respectively. Following this, the sum of $Na^+$, $K^+$ and $Mg^{2+}$ in meq/L is corrected for the seawater contribution—specified from the $Cl^-$ content. The cation exchange code is given following the sign of the cation exchange parameter (Na + K + Mg) corrected, with a margin of error of $\sqrt{0.5Cl}$ (negative code = deficit, positive code = surplus, $\theta$ code − within margin error = equilibrium). The $Cl^-$ content is assumed to be a conservative component originating from seawater only.

#### 3.3.2. Hydrochemical Facies Using Piper and HFE-D Diagrams

The Piper diagram plots the main cations and anions using relative concentration expressed in meq/L in a multivariate diagram including a cation triangle and an anion triangle and a transferred rhomb. The latter are also indicators of salinization/freshening processes caused by cation exchange. The water type of the first fresh-water end-member is CaHCO$_3$0 and the salt water end-member is NaCl0. The facies of the samples shifting from CaHCO$_3$0 → CaCl$^-$ → NaCl$^-$ → NaCl0 on and above the mixing line present the salinization process with "-" expressing a deficit of marine cations. On the other hand, the samples located below the mixing line are experiencing freshening processes and the facies of the waters vary from NaCl0 → NaCl$^+$ → NaHCO$_3^+$ → MgHCO$_3^+$ → CaHCO$_3^+$ → CaHCO$_3$0 with the "+" describing a surplus of marine cations.

The alternating sequence between freshening and salinization processes can be recognized by the evolution of the chemistry of the major ions over time [55] used a Hydrochemical Facies Evolution Diagram (HFE-Diagram) to describe the seawater intrusion based on the evolution of hydrochemical facies. The HFE-Diagram is plotted in terms of the relative percentages of anions and cations, in which $Cl^-$ represents the seawater end-member, whereas $HCO_3^-$ or $SO_4^{2-}$ corresponds to the dominant anion in fresh groundwater [54]. The base-exchange reactions were described following the percentages of $(Na^+ + K^+)$ and $Ca^{2+}$. The blue mixing line is situated between the FW/SW end-members. Besides the mixing line, there are two transitions indicating the evolution of water samples according to freshening and saltwater intrusion processes. Different sub-stages can be identified, using the salinity evolution estimated through the proportion of $Cl^-$: in the freshwater intrusion (FwI) phase, represented by the top left curved arrow, the freshening sub-stages (f1, f2, f3, f4 and FW) can be identified. In the opposite sector of the diagram, the saltwater intrusion (SwI) sub-stages belonging to intrusion phases are represented (i1, i2, i3, i4 and SW).

### 3.3.3. Origin of Solutes Using Conservative Element Behavior, Aqueous Speciation and Mineral Saturation Indices

The origin of solutes and their mixing were assessed using the relationship of $Cl^-$ and other ions. $Cl^-$ and $Br^-$ were considered as the characteristic saline elements because of their conservative behavior with their hydrophilic character and small ionic size. They neither take part in significant ion-exchange reactions at low temperatures nor are adsorbed onto mineral surfaces. As they are highly soluble, they only form minerals during extreme evaporation conditions when halite starts to precipitate [56]. The major ions were plotted against $Cl^-$ content and compared with the mixing line to inspect whether ions were enriched or depleted. The calculations of aqueous speciation and mineral saturation indices were carried out using the aqueous geochemical modelling code PHREEQC [57]. For these calculations, the saturation index of calcite, dolomite, gypsum and halite for each water sample were examined to see if specific rock/water interaction had occurred.

### 3.3.4. Expected Composition Based on Conservative Mixing

The chemical interactions between fresh and saltwater were detected by calculating the expected composition based on conservative mixing of saltwater and freshwater, and then comparing the result with the measured concentrations in the water sample, $Cl^-$ being considered as a conservative component. Assuming no $Cl^-$ component came from the dissolution of halite, the fraction of seawater $f_{sea}$ was calculated using the $Cl^-$ concentration of the sample [19,39] following Equation (1):

$$f{sea} = \frac{m_{Cl^-,sample} - m_{Cl^-,fresh}}{m_{Cl^-,sea} - m_{Cl^-,fresh}}, \tag{1}$$

where $m_{Cl-sea}$ and $m_{Cl-fresh}$ were taken from the seawater composition from [58] and freshwater from Luy river catchment with the lowest $Na^+$ and $Cl^-$ contents, respectively.

The concentration of ion i, by conservative mixing of seawater and freshwater, was then calculated by Equation (2):

$$m_{i,\,mix} = f_{sea} \times m_{i,\,sea} + (1 - f_{sea}) \times m_{i,\,fresh}, \tag{2}$$

where $m_i$ is the concentration of i (mmol/L), $f_{sea}$ the fraction of seawater in the mixed water and subscripts "*mix*", "*sea*" and "*fresh*" indicate a conservative mixture, seawater and freshwater, respectively. During the reactions, any change of the concentration $m_{i,\,react}$ was simply determined by Equation (3):

$$m_{i,\,react} = m_{i,\,sample} - m_{i,\,mix}, \tag{3}$$

where $m_{i,\,sample}$ is the measured concentration in the sample.

If $m_{i,\,react}$ is greater than zero, it indicates the groundwater was enriched with respect to species i, whereas a $m_{i,\,react}$ lower than zero suggests depletion compared to plain mixing.

## 4. Results

### 4.1. Chemical Composition and Ion Distribution of Water Samples

Except for four samples taken from wells drilled through the bedrock, which is located between 20 and 50 m below the surface except in the upstream part of the study area, samples were collected from the shallow unconsolidated aquifers.

A statistical summary of the hydrochemical parameters is presented in Table 1. Minimum and maximum values of the TDS are 985 and 16,155 mg/L respectively, with an average value of 1617 mg/L, illustrating the large variability in salinity. The pH ranges in the expected interval for groundwater from 4.2 to 8.7.

**Table 1.** Basic statistical descriptors of the physicochemical parameters from groundwater and surface water: n—number of samples, max—maximum, min—minimum, med—median, ave—average and std.—standard deviation. Units: Temperature (Temp) in °C, EC in µS/cm (25 °C), total dissolved solids (TDS) and ions in mg/L.

| | pH | EC | TDS | Na$^+$ | K$^+$ | Ca$^{2+}$ | Mg$^{2+}$ | FeTotal | Mn$^{2+}$ | NH$_4^+$ | Cl$^-$ | SO$_4^{2-}$ | NO$_3^-$ | HCO$_3^-$ | PO$_4^{3-}$ | Br$^-$ |
|---|---|---|---|---|---|---|---|---|---|---|---|---|---|---|---|---|
| | | | | | | | Depth from 3.5 to 110 m | | | | | | | | | |
| N | 110 | 110 | 110 | 110 | 110 | 110 | 110 | 110 | 110 | 110 | 110 | 110 | 110 | 110 | 110 | 110 |
| Max | 8.7 | 35,509 | 16,155 | 4903 | 150 | 979 | 912 | 31 | 23 | 15 | 4935 | 4219 | 341 | 1022 | 20 | 108 |
| Min | 4.2 | 162.0 | 85.6 | 8.9 | 0.3 | 3.3 | 2.0 | 0.0 | 0.0 | 0.0 | 16.2 | 0.0 | 0.2 | 0.0 | 0.0 | 0.3 |
| Med | 7.3 | 1471.5 | 984.6 | 173.7 | 5.6 | 53.0 | 25.9 | 0.0 | 0.2 | 0.1 | 138.4 | 104.6 | 22.3 | 366.0 | 0.3 | 2.5 |
| Ave | 7.2 | 2968.9 | 1617.1 | 358.9 | 20.6 | 85.3 | 51.6 | 1.5 | 0.8 | 0.8 | 425.1 | 210.3 | 48.7 | 404.0 | 0.9 | 7.7 |
| Std | 0.68 | 4856.1 | 2055.7 | 622.8 | 30.7 | 130 | 110.3 | 4.5 | 2.4 | 2.1 | 877.8 | 444.6 | 61.8 | 239.4 | 2.3 | 16.9 |
| | | | | | | | WHO standards of maximum value for drinking water (mg/L) | | | | | | | | | |
| | | | 600 | 200 | | | 50 | | | | 250 | 250 | 50 | | | |
| | | | | | | | Degree of Restriction on Use for irrigation | | | | | | | | | |
| | | | mL | meq/L | | | | | | | meq/L | | mg/L | mg/L | | |
| Slight to Moderate | | | 450–2000 | >3 | | | | | | | 4–10 | | 5–30 | 1.5–8.5 | | |
| Severe | | | >2000 | >9 | | | | | | | >10 | | > 30 | >8.5 | | |

Na$^+$ is the most abundant cation in the water samples with concentration ranging from 10 to 4902 mg/L. Ca$^{2+}$ ranges from 3.2 to 979 mg/L and Mg$^{2+}$ from 1.9 to 912 mg/L. The highest value of K$^+$ is 150 mg/L and the lowest one is 0.25 mg/L.

For the anions, the concentration of Cl$^-$ varies from 16.2 to 4935 mg/L and SO$_4^{2-}$ ranges from 0 to 4219 mg/L. This largest value, which is higher than the SO$_4^{2-}$ concentration from seawater (2750 mg/L), was collected from LK07-BT, likely indicating another source of contamination. NO$_3^-$ concentration varies from 0.2 to 340 mg/L and PO$_4^{3-}$ concentration from 0 to 19 mg/L.

Figure 4 shows the well screen position and the contour plots of the TDS value, Cl$^-$, Na$^+$, Ca$^{2+}$, Mg$^{2+}$, SO$_4^{2-}$, HCO$_3^-$, NO$_3^-$ and Br$^-$ distribution along the transect DC (see Figure 1), using samples located on or in the direct vicinity (less than 1 km) of the section line.

Despite the lack of samples located below −10 m, the TDS, Cl$^-$, Na$^+$, Mg$^{2+}$ and Br$^-$ and to a lesser extent SO$_4^{2-}$ show similar patterns and constitute evidence of saltwater presence in the lower aquifer as all the samples collected just above the bedrock show the highest values.

The distribution of HCO$_3^-$ in Figure 4h is quite different and likely brings information regarding freshwater recharge. The Ca$^{2+}$ concentration has a mix pattern, showing influence of both freshwater recharge and salinization.

The NO$_3^-$ concentration is large (greater than 50 mg/L) in the upper aquifer, showing that groundwater is polluted, likely due to the extensive use of fertilizers for agriculture. The concentration decreases in deeper samples, as expected from natural nitrate reduction.

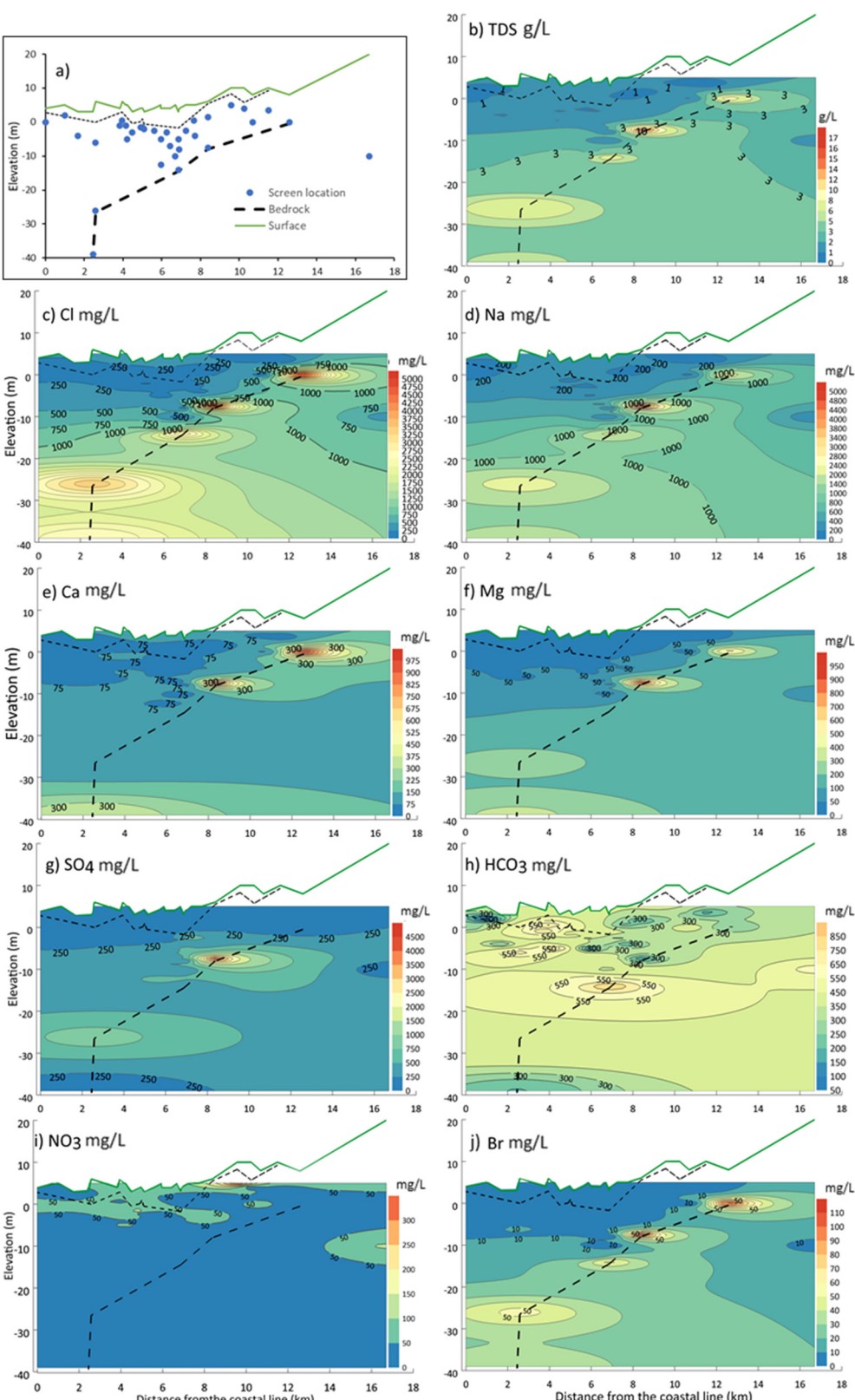

**Figure 4.** (**a**) Vertical well screen positions along the CD profile in Figure 1. The thick black dashed line indicates the bedrock surface, the green line is the surface elevation line and the thin black line is the elevation of the water table in the wells. (**b**–**j**) interpolation of TDS and ion concentration from water samples along profile CD.

Among the 110 water samples, 72 water samples are exceeding the WHO limit [59] for drinking water (Table 1) for at least one ion, including $Mg^{2+}$—18 samples, $Na^+$—48 samples, $Cl^-$—33 samples, $SO_4^{2-}$—24 samples, $NO_3^-$—41 samples and TDS—53 samples. Following the guidelines for evaluation of water quality for irrigation given by Ayers and Westcot [60] (Table 1), all water samples have a degree of restriction on use for irrigation including TDS—82 samples, $Na^+$—46 samples, $Cl^-$—30 samples, $NO_3^-$—32 samples and $HCO_3^-$—70 samples for the slight–moderate degree; and TDS—19 samples, $Na^+$—48 samples, $Cl^-$—24 samples, $NO_3^-$—46 samples and $HCO_3^-$—32 samples for the severe degree.

*4.2. Classification of Water*

According to Stuyfzand's classification, 56% of the water samples are fresh whereas 19% and 15% are fresh-brackish and brackish, respectively. Only 10% of the samples in the Luy river catchment are brackish-salt. The spatial distribution of water types in terms of $Cl^-$ content is shown in Figure 5a. Most brackish-salt water samples are located along the Luy river close to the sea and at larger depth. However, some brackish and brackish-salt samples taken at depth less than 15 m are located far from the coast up to 15 km inland. Globally, there is no clear trend in the distribution of salinity in shallow samples. This observation confirms geophysical observations that identified a strong variability at the small scale [34]. Samples collected in the pairs of shallow/deep wells also systematically indicate a higher salinity at depth.

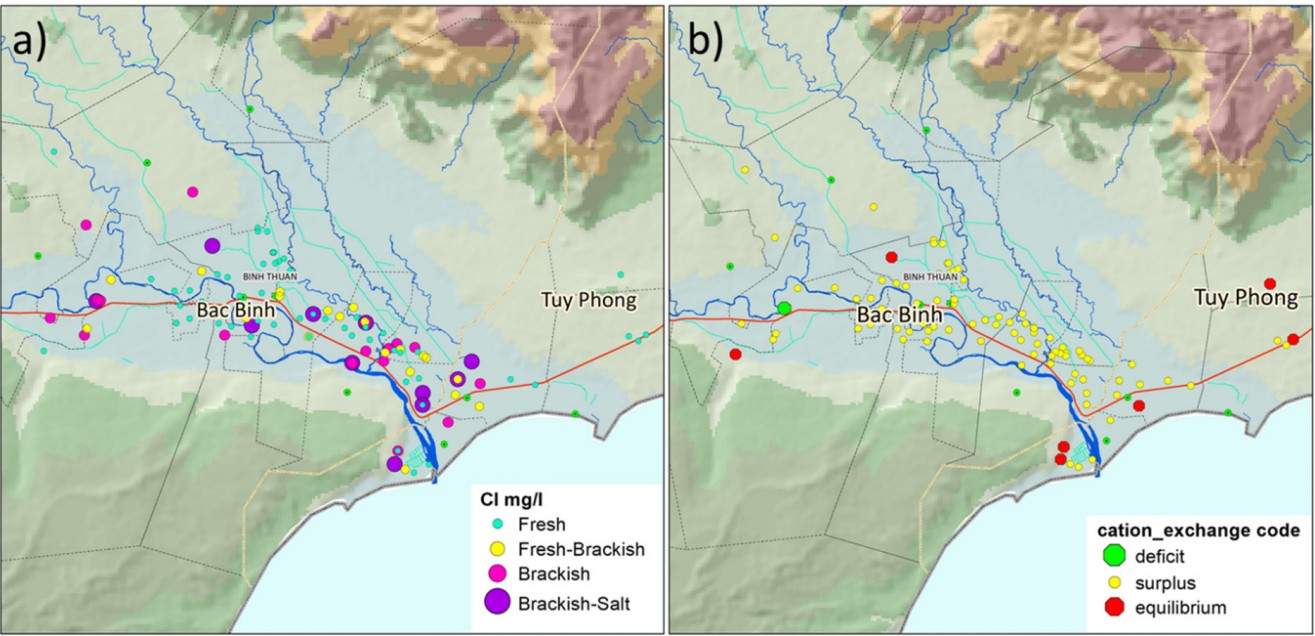

**Figure 5.** (**a**) Water type distribution based on the $Cl^-$ content and (**b**) the distribution of cation exchange code of water samples in the Luy river catchment.

Figure 5b illustrates the spatial distribution of the cation exchange code. Most samples in the Luy river catchment show a surplus of (Na + K + Mg), indicating that freshening processes are dominant. This can be counterintuitive as the local population is experiencing salinization of its water resources [61]. However, freshening processes are generally acting at a longer time scale than salinization [21]. The hydrochemical composition of water needs more time to respond to a new equilibrium compared to hydrodynamics. Therefore, from the observations, it seems that most of the samples in the Luy River have been experiencing freshening processes in the past, even if salinization processes might take place in the present.

Most of the samples in equilibrium are located in the dunes on the sides of the Luy river; only one sample is located in the plain. This suggests that the chemical composition

of groundwater has reached an equilibrium in these locations, which can be related to a preferential recharge in dune areas [62]. Only one sample in deficit is observed, located far away from the coast.

### 4.3. Hydrochemical Facies Using Piper and HFE-D Diagrams

The Piper diagram in Figure 6a shows the groundwater composition and water types. The NaCl water type is the most abundant (44.5% of the samples) whereas the Na-Mg-SO$_4$ water type represents the minority (1.8% of the samples). The mixing Na-HCO$_3$-Cl water type corresponds to 20.9% and the Ca-Mg-HCO$_3$, Ca-Mg-Cl-SO$_4$ and Na-HCO$_3$ water types represent 15, 13 and 5% of the samples, respectively.

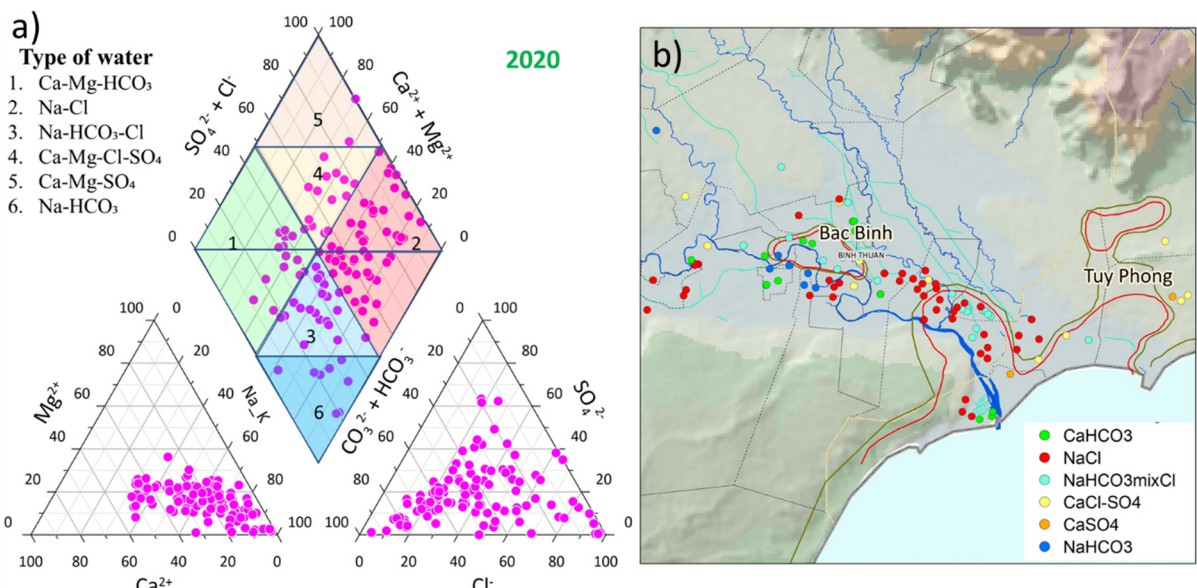

**Figure 6.** (**a**) Piper diagrams of groundwater samples in the Luy river catchment (the purple balls) and (**b**) water type from the Piper diagram distribution.

The distribution of the water types from the Piper diagram is mapped in Figure 6b. The NaCl water type extends along the Luy river from the coastal line to around 18 km inland. The distribution of groundwater samples in the Piper diagrams shows that the freshening process is dominant.

Figure 7a illustrates the distribution of groundwater samples in the HFE diagram. The 8 sub-stages of freshening and saltwater intrusion combined with their water type are mapped in Figure 7b and listed in Table 2. The map reveals that the freshening phase extends largely in the study area, which is characterized by the sub-stages of proximal facies f4 + FW and f3. This can be explained by the abundance of the shallow water samples and the lack of deep-water samples. The distal facies corresponding to sub-stages f2 and f1 are less abundant and some occurrences are observed far from the coast. The main recharge areas are recognized by the freshening facies that are most proximal to the FW end-member observed on the right bank of the Luy river and at the higher elevation parts in the left bank.

SI is observed in the coastal zone with the proximal facies i3 and SW + i4 at different depths in both deep and shallow wells. These samples might result directly from the sea or the estuary. Both the proximal and distal saltwater intrusion facies are also observed locally far inland, mostly from shallow wells of less than 15 m depth. The saltwater and salinization processes in the shallow boreholes (less than 15 m depth) might be explained either by infiltration from the river, when close to it, by upconing processes from the deeper aquifer when the shallow aquifer is exploited or by the existence of clay lenses where saltwater has been trapped.

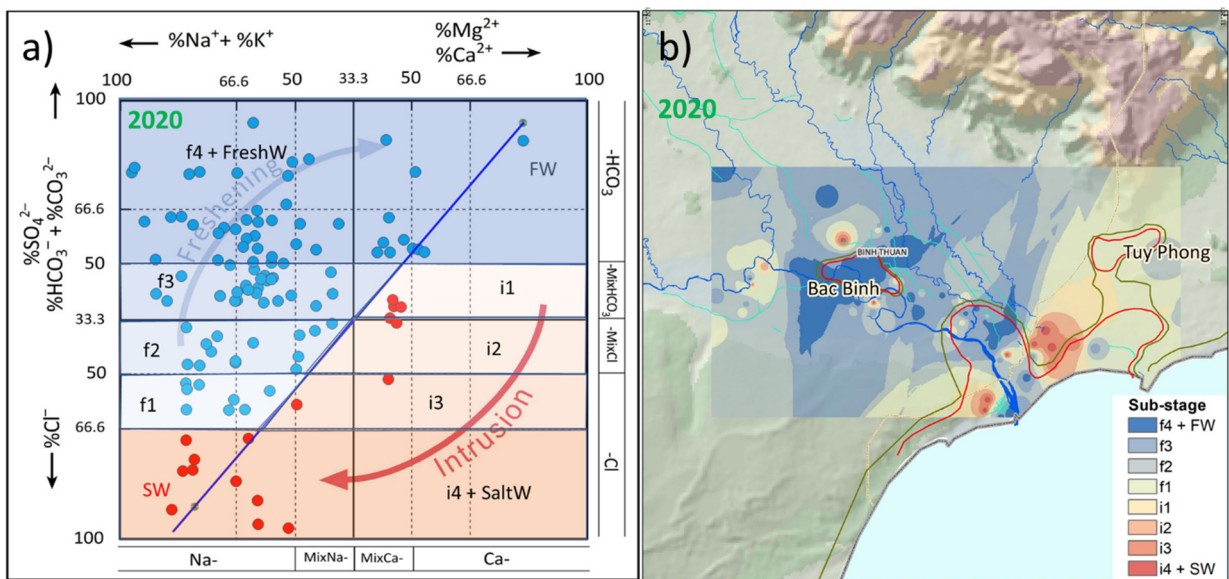

**Figure 7.** (**a**) HFE diagram and (**b**) distribution of sub-stages of water samples in the Luy river catchment, the red balls are the samples in the intrusion zone and the blue balls are in the freshening zone.

**Table 2.** Distribution of sub-stages in freshening and intrusion phases for 110 water samples collected in the rain season 2020.

|  | FW + f4 | f3 | f2 | f1 | i1 | i2 | i3 | SW + i4 | Total |
|---|---|---|---|---|---|---|---|---|---|
| Number | 45 | 26 | 14 | 8 | 4 | 1 | 2 | 1 | 110 |
| Percentage | 41 | 24 | 13 | 7 | 3.5 | 1 | 1.5 | 1 | 100 |
|  | freshening | | | | intrusion | | | | |
|  | proximal ← distal | | | | distal → proximal | | | | |

### 4.4. Ionic Relation and Origin of Solutes

The composition of groundwater samples was plotted using pairs of ions (Figure 8). Figure 8a,b show the relation of Na$^+$ versus Cl$^-$; the halite dissolution was added as the 1:1 line. Samples lying on the mixing line or 1:1 line indicate simple mixing processes and halite dissolution, respectively. Most of the samples observed above the mixing line and 1:1 line show an Na$^+$ surplus that might originate from cation exchange in freshening processes. The Cl$^-$ vs. Na$^+$/Cl$^-$ ratio diagram shows that almost all samples are located above the seawater ratio line, confirming an excess of Na$^+$, with a few samples located on the mixing line.

Figure 8c shows the correlation of Ca$^{2+}$ versus Cl$^-$. Many samples located above the theoretical mixing line indicate an excess of Ca$^{2+}$. Having a majority of samples with an excess is unexpected, as we would expect a depletion resulting from cation exchange of the freshening process in which Ca$^{2+}$ from freshwater is absorbed and Na$^+$ is released into the groundwater. This enrichment of Ca$^{2+}$ may thus come from calcite/dolomite dissolution or freshwater recharge. This will be further investigated using the saturation index.

Figure 8d shows the relation of Ca$^{2+}$ versus HCO$_3^-$ adding the calcite dissolution line. The high HCO$_3^-$ value indicates the freshwater environment and the bicarbonate enrichment. The much higher HCO$_3^-$ content (a thousand mg/L) combined with the excess of Ca$^{2+}$ might be related to the second stage of calcite dissolution.

The SO$_4^{2-}$ anion is considered to originate primarily from seawater. The surplus of sulfate in the SO$_4^{2-}$ vs. Cl$^-$ diagram (Figure 8e) where most of the water samples are located above the mixing line suggests not only a seawater origin but also another contamination source, such as fertilizers.

The plot of $NO_3^-$ vs. $Cl^-$ in Figure 8f shows the enrichment of nitrate up to 5.5 meq/L. The excess of nitrate might originate from the infiltration of fertilizers from the surface as agriculture activities are intense in the Luy river catchment.

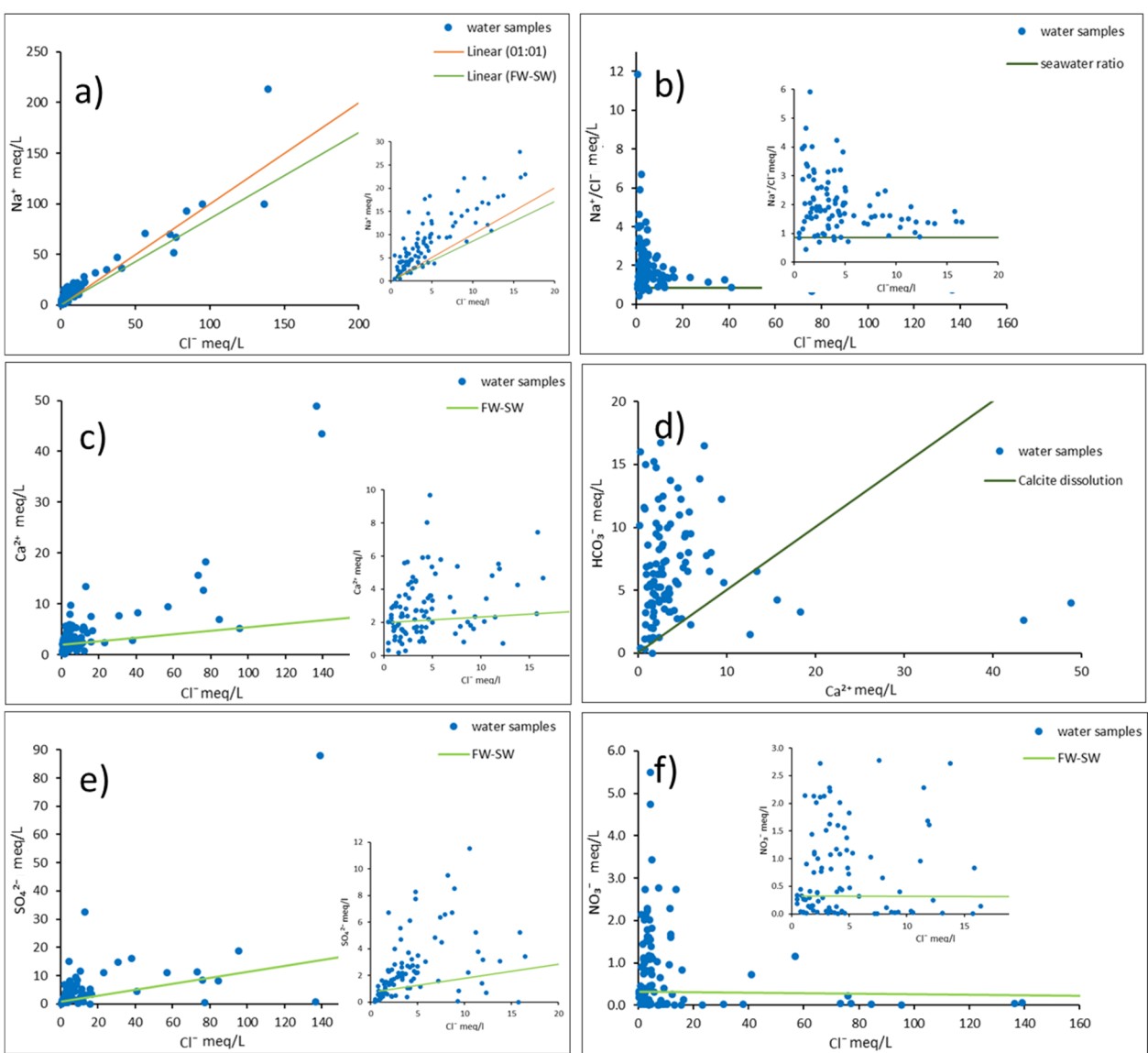

**Figure 8.** Plots of $Na^+$ (**a**), $Ca^{2+}$ (**c**), $HCO_3^-$ (**d**), $SO_4^{2-}$ (**e**) and $NO_3^-$ (**f**) with respect to $Cl^-$ and their ionic ratios (**b**) for groundwater samples.

### 4.5. Water/Rock Interaction and the Origin of Solutes

The Luy river plain is characterized by shallow Holocene sediments including clay, silt, sand mixed with gravels, grits and pebbles or shelf fragments originating from fluvial-marine sediments. The deeper sediments consist of fine to medium grain quartz sand from the Pleistocene. The bedrock mainly consists of granite, sandstone and siltstone, which also surrounds the basin in its upstream part.

If $Ca^{2+}$, $Mg^{2+}$, $SO_4^{2-}$ and $HCO_3^-$ are resulting from calcite, dolomite and gypsum dissolution, then an equilibrium should exist between cations and anions [63]. It also means water samples should be in or close to the 1:1 linear trend.

The cation exchange factor plays a significant role in the freshening and salinization process [21]. In a saturated solution, $Ca^{2+}$ and $Mg^{2+}$ supplied by calcite, dolomite and gypsum dissolution are replaced by $Na^+$ originating from NaCl. On the other hand, $HCO_3^-$ presents the calcite and dolomite dissolution and $Cl^-$ and $SO_4^{2-}$ record how much halite

and gypsum are dissolved, respectively. A diagram of $(Ca + Mg - SO_4 - HCO_3)$ versus $(Na + K - Cl)$ (Figure 9a) is plotted to examine the amount of $Na^+$ gained or lost relative to the supply by halite dissolution and the gain and loss of $Ca^{2+}$ and $Mg^{2+}$ related to the supplement by gypsum, calcite and dolomite dissolution. The linear correlation of these parameters with a slope of $-1$ suggests (Figure 9b) the cation exchange governs the composition of groundwater, and the lower right corner in the diagram represents the participation of cation exchange in the freshening process [63].

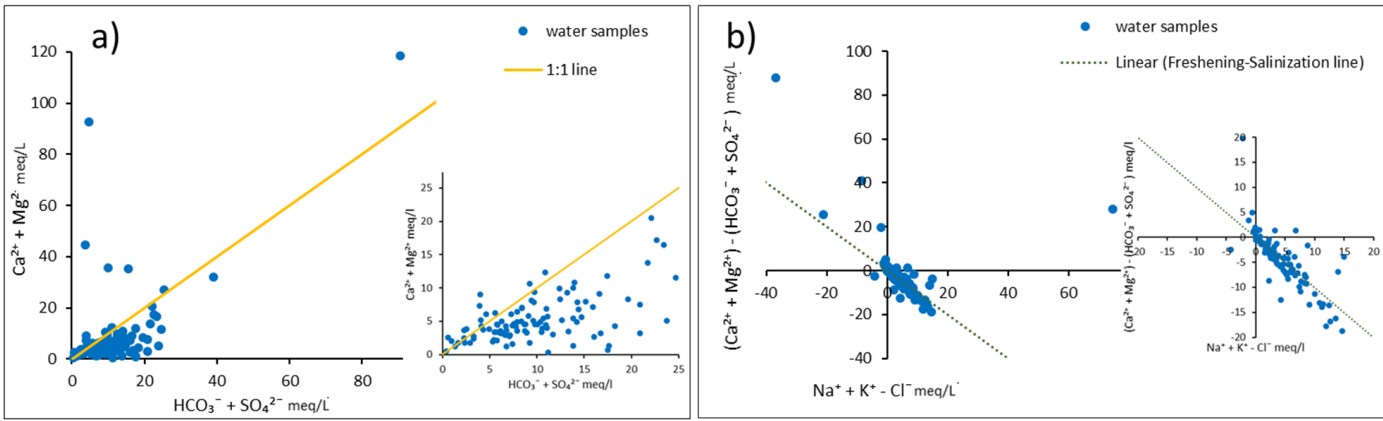

**Figure 9.** Relationship between $(HCO_3 + SO_4)$ versus $(Ca + Mg)$ (**a**) and $(Ca + Mg - SO_4 - HCO_3)$ versus $(Na + K - Cl)$ (**b**).

The water samples are located mostly under the 1:1 line, presenting an enrichment of $HCO_3^- + SO_4^{2-}$ and a deficiency of $Ca^{2+} + Mg^{2+}$, then the residual negative charge of $HCO_3^- + SO_4^{2-}$ must be balanced by $Na^+$. This suggests cation exchange processes are occurring during freshening and confirms the analysis of the processes of freshening and saltwater intrusion from other analysis.

### 4.6. Speciation-Solubility Calculations

The most common forms of calcium in sedimentary rock are carbonates. There are two crystalline forms, calcite and aragonite, with the same formula $CaCO_3$, and the mineral dolomite can be described as $CaMg(CO_3)_2$. Other calcium minerals common in sediments consist of the sulfates gypsum ($CaSO_4.2H_2O$) and anhydrite ($CaSO_4$) and more rarely, fluorite ($CaF_2$). Calcium can also exist as a component of some types of zeolites and montmorillonite. The dissolution of the retained shells can be simply represented as the ion $Ca^{2+}$ in the solution.

According to Hem (1985) [64], in some conditions, a bicarbonate concentration near 1000 mg/L corresponds to about 10% of the calcium found in $CaHCO_3^+$ form. Similarly, more than half of the calcium might be represented in the form of the $CaSO_4$ ion pair if sulfate concentrations are above 1000 mg/L.

The saturation index of calcite, dolomite, halite and gypsum is plotted in Figure 10 against the concentration in $Cl^-$, used as a proxy for salinity. Fifty-six percent of the water samples represented in Figure 10a are supersaturated with respect to calcite. Such an index value reflects groundwater containing an abundant amount of calcite with sufficient resident time to reach equilibrium; calcite will be precipitated in suitable conditions. The rest of the samples are undersaturated with respect to calcite, suggesting the samples might come from an environment poor in calcite or where $Ca^{2+}$ has another form or has not reached equilibrium due to short residence time. Figure 10b shows that dolomite has a similar trend with 57% oversaturated samples with dolomite respectively reflecting a dolomite enriched environment.

The principal source of carbon dioxide species that produce alkalinity in surface or groundwater is the $CO_2$ gas fraction of the atmosphere or the atmospheric gases present

in the soil or the unsaturated zone lying between the surface of the land and the water table [63]. When $CO_2$ dissolves in water, gaseous $CO_2(g)$ becomes aqueous $CO_2$ (aq) and forms carbonic acid, $H_2CO_3$ that can favor calcite dissolution.

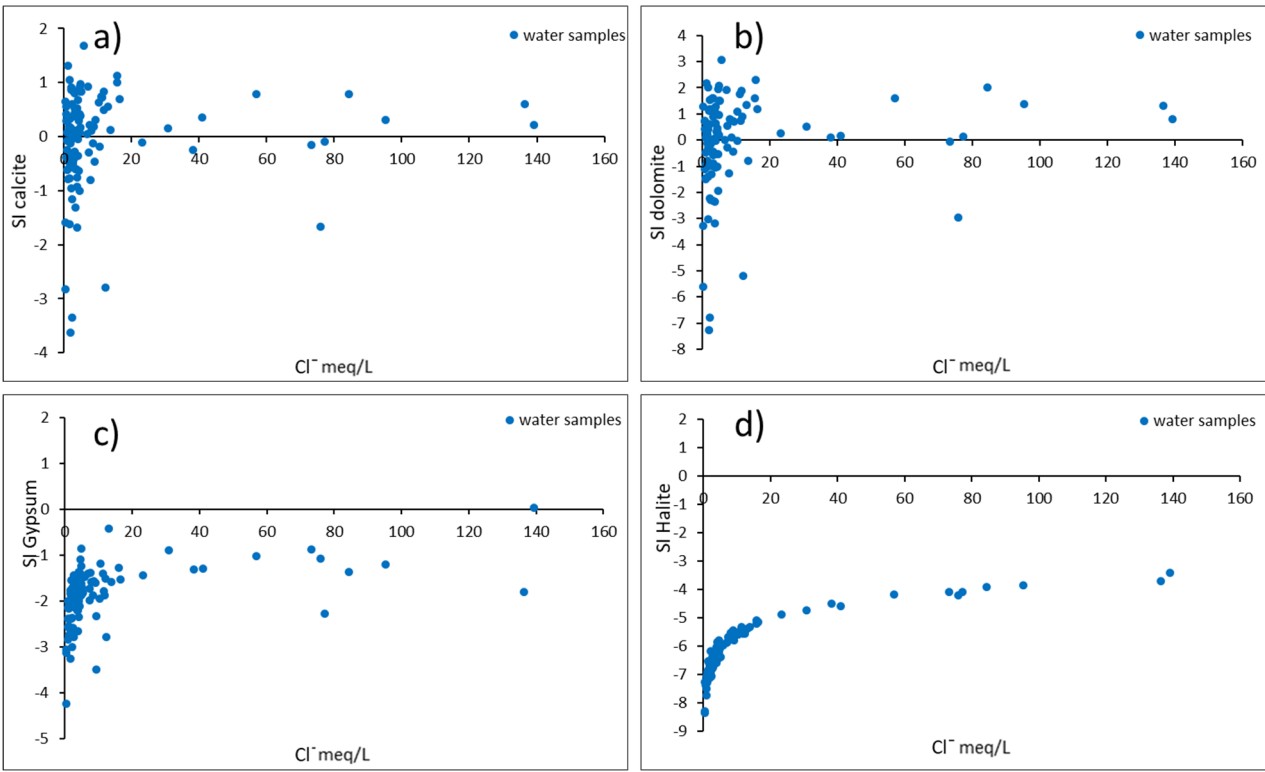

**Figure 10.** Saturation index for calcite (**a**), dolomite (**b**), gypsum (**c**) and halite (**d**) versus $Cl^-$ concentration.

Fifty-six percent of water samples have a positive saturation index with respect to calcite and dolomite. Because groundwater samples collected at a depth lower than 10 m are abundant, they might have an interaction with $CO_2(g)$. Then, the $HCO_3^-$ concentration could increase in the water samples following the dolomite and calcite dissolution reactions. It also causes higher $Ca^{2+}$ and $Mg^{2+}$ concentrations which will be absorbed by clay and desorbed $Na^+$ in the water samples. This process can explain the high $HCO_3^-$ concentration and the apparent enrichment of $Ca^{2+}$ (Figure 8c). Therefore, a high $HCO_3^-$ concentration results from either $CO_2(g)$ dissolution in the groundwater samples at the pH variation between 6.3 and 10.3 or from being produced from calcite and dolomite dissolution at a pH less than 6.3.

The water samples are all undersaturated with respect to halite, suggesting there are no minerals formed of halite in the study area. The same observation is valid for gypsum except for one oversaturated sample which may relate to the dissolution of gypsum from previous saltwater intrusion [65].

An ionic delta was calculated for all major cations $Na^+$, $Ca^{2+}$, $Mg^{2+}$ and $K^+$ and anions $HCO_3^-$, $SO_4^{2-}$ and $NO_3^-$. The result confirms the cation exchange during the freshening process, as well as the enrichment of the anions $SO_4^{2-}$ and $NO_3^-$ in some samples, indicating pollution.

### 4.7. Variation of Hydrogeological Parameters of Groundwater with Time

The sampling campaign was repeated during the dry season of 2021. We could only record information from 90 out of the 110 wells visited in 2020, the other ones were either dry or had been abandoned. The latter often signifies that the salinity has increased to a level unacceptable for irrigation. Only 20 wells had lower TDS values from 15 to 4821 mg/L,

whereas 70 wells had higher TDS from 7 to 32,054 mg/L. Figure 11 shows the locations of the wells with increasing and decreasing salinity as well as the water level going up or down between the two measuring periods. In general, the salinity increased, mostly in the wells located in the left bank and decreased in the right bank of the Luy river. Water level seems to go down in the inland part and rise up in the estuary area, which might be a result of the tides. Based on these results, 33 groundwater samples were collected, including the 18 water samples from our new monitoring wells (except the two that were dry) and the 15 samples that had an absolute difference in TDS value greater than 500 mg/L.

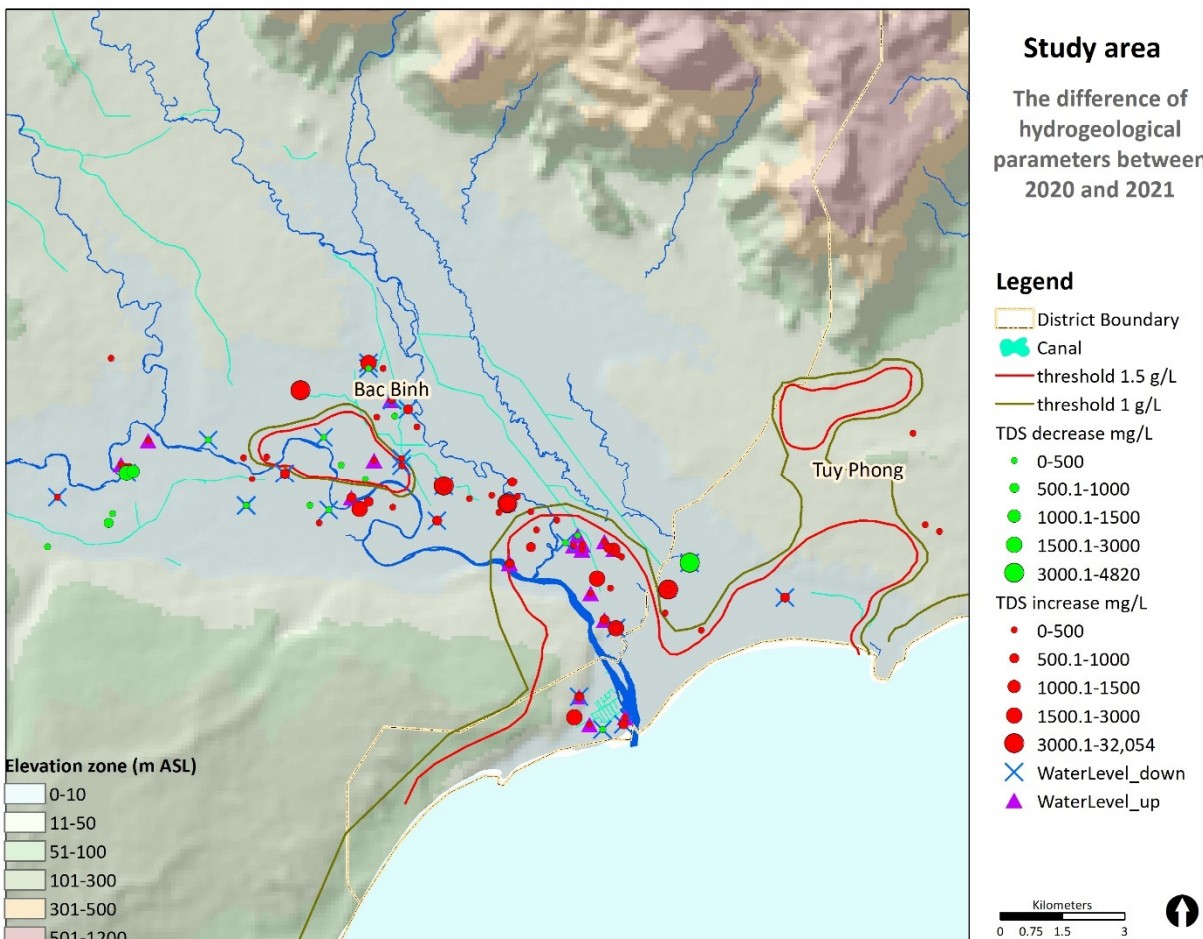

**Figure 11.** The difference of the hydrogeological parameters between 2020 and 2021.

The 33 water samples collected in both the rain season 2020 and dry season 2021 were used to analyze their temporal evolution (Table 3). On average, the concentration of all the ions increased except for $PO_4^{3-}$ (stable) and $NO_3^{-}$ (decreasing trend). The ions $Na^+$, $Cl^-$ and $SO_4^{2-}$ present extremely high concentrations at some locations in 2021 compared to 2020 (such as $Na^+$—8204 mg/L, $Cl^-$—19,752 mg/L and $SO_4^{2-}$—5865 mg/L) that identify the presence of saltwater in the groundwater samples ($Cl_{seawater}$ = 19,000 mg/L) and a possible stronger SI in the dry season.

Similar patterns of increasing salinity in the dry season are evident in the primary type classification because the fresh and fresh brackish water types decrease from 57.6 to 30% in total whereas the brackish water and salt water types increase from 12 to 27% and from 0 to 6%, respectively. The brackish-salt water type decreases slightly from 30 to 27%. This suggests the salinity increases in both freshwater and saline water zones in the dry season.

The Stuyfzand classification in terms of cation exchange code shows that the water samples which have a surplus code corresponding to water samples involved in the freshening process (release of $Na^+$) decrease from 28 samples (85%) to 22 samples (67%).

In contrast, the samples in deficit which indicate saline intrusion with cation exchange processes increase from one sample (3%) to seven samples (21%). Four samples (12%) are in equilibrium in both sampling periods but not necessarily at the same locations. This is a confirmation of the increased salinity in the dry season. Figure 12 shows the location of the samples with their cation exchange codes changing between 2020 and 2021. The deficit code (indicating saltwater intrusion) is visible up to 14 km from the coast. This identifies the presence of saltwater intrusion at a larger distance from the river in the dry season of 2021. The salinization process can hardly be explained with direct intrusion from the river but is likely related to pumping activities inducing some upconing from saltwater already present at depth or trapped in clay-rich zones.

**Table 3.** Summary of chemical composition variation of 33 water samples.

| mg/L | TDS | $Na^+$ | $K^+$ | $Ca^{2+}$ | $Mg^{2+}$ | $Cl^-$ | $SO_4^{2-}$ | $NO_3^-$ | $HCO_3^-$ | $PO_4^{3-}$ | $Br^-$ |
|---|---|---|---|---|---|---|---|---|---|---|---|
| | | | | Increasing from 2020 to 2021 in 33 samples | | | | | | | |
| % | 88 | 94 | 82 | 76 | 79 | 91 | 61 | 36 | 58 | 45 | 76 |
| Max | 23,729 | 5921 | 303 | 306 | 1157 | 16,372 | 1646 | 173 | 246 | 1 | 152 |
| Min | 3.7 | 0.4 | 0.1 | 1.3 | 0.1 | 0.8 | 2.2 | 0.4 | 3.7 | 0.1 | 0.2 |
| Ave | | 453 | 22 | 69 | 93 | 1346 | 204 | 640 | 718 | 0.3 | 11 |
| | | | | Decreasing from 2020 to 2021 in 33 samples | | | | | | | |
| % | 12 | 6 | 18 | 24 | 21 | 9 | 39 | 64 | 42 | 21 | 21 |
| Max | 2056 | 512 | 23 | 129 | 12 | 990 | 621 | 36 | 267 | 4 | 12 |
| Min | 63 | 33 | 0.2 | 0.5 | 0.3 | 0.4 | 2.3 | 0.7 | 8.5 | 0.1 | 0.5 |
| Ave | 623 | 273 | 6 | 38 | 5 | 332 | 111 | 7 | 112 | 1 | 5 |

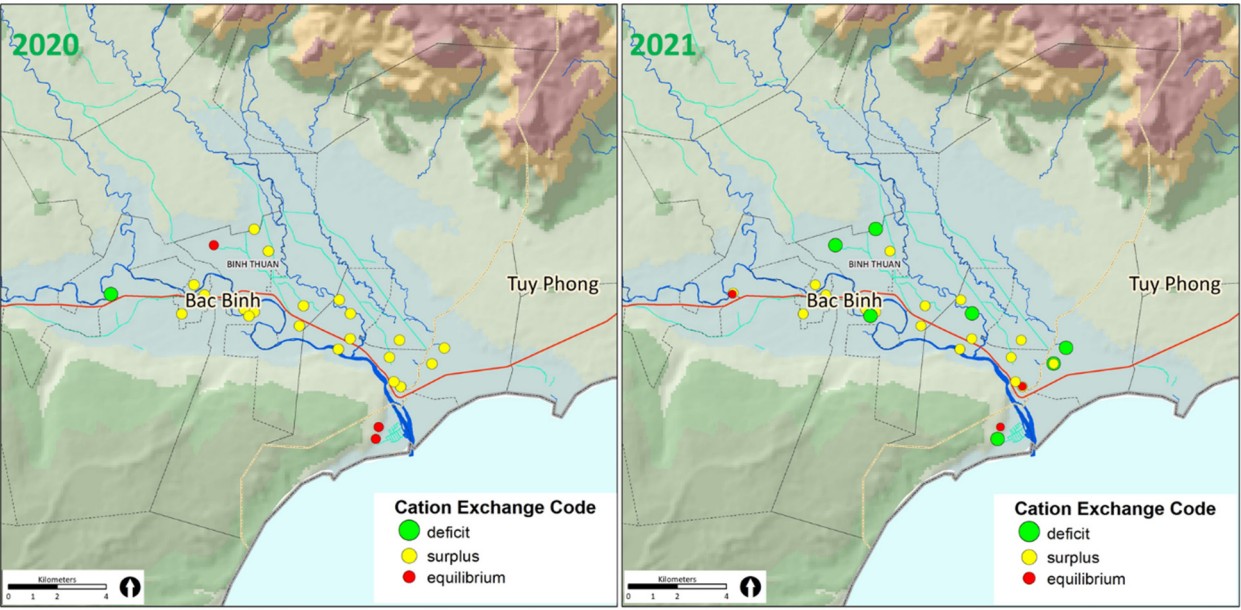

**Figure 12.** Distribution of the cation exchange codes of 33 water samples collected in 2020 and 2021.

The water samples collected in 2020 and 2021 at the same locations are plotted in the HFE diagram to observe the water faces variation during the time periods (Figure 13).

The distribution of sub-stages in freshening and intrusion phases for the two sampling periods is shown in Table 4. Water samples situated in the freshening field decrease from 73% in 2020 to 60% in 2021. The sub-stages range regularly from distal to proximal to the freshwater in the freshening field whereas there are few samples located in the distal and proximal facies in the intrusion field. The water samples located in the intrusion field at the two periods are mostly located in the seawater field which is located in the first quarter

corner of the HFE-Diagrams. This representation identifies that freshening is the main process occurring in this study area, resulting in an increase in the marine cations $Na^+$, $K^+$ and $Mg^{2+}$ and decrease in $Ca^{2+}$ in the groundwater, but that seasonal variations are impacting the freshening sub-stages.

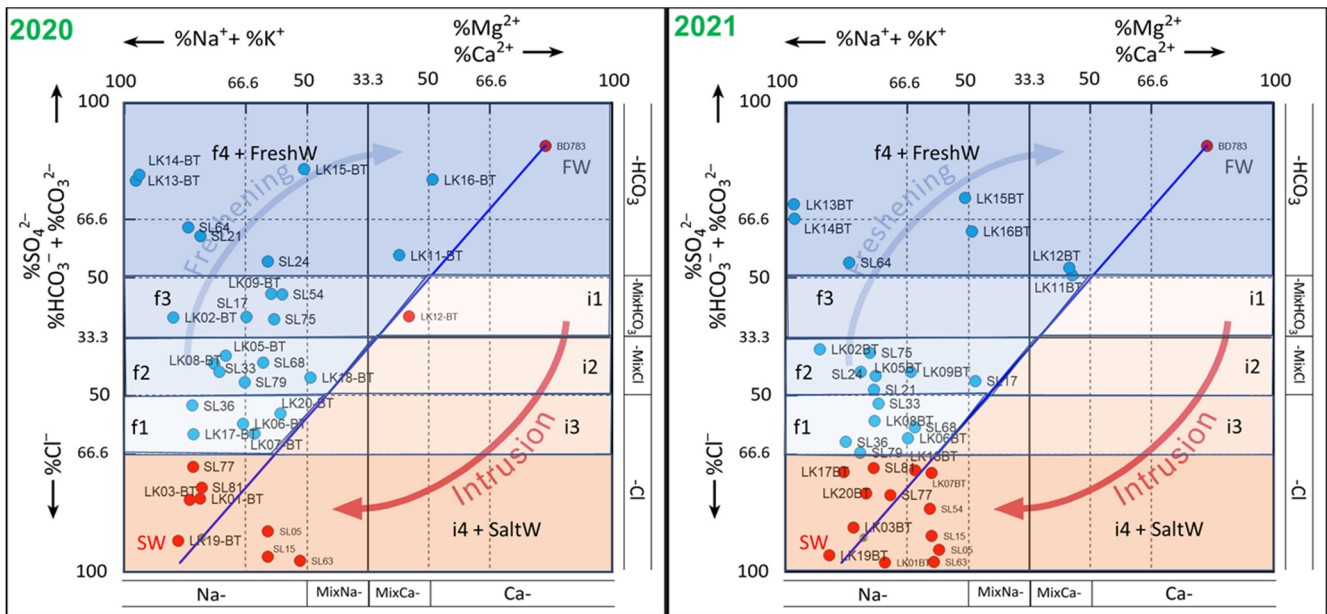

**Figure 13.** Representation of 33 samples collected in the rain season 2020 and dry seasons 2021 on the HFE-Diagram.

**Table 4.** Distribution of sub-stages (in %) in freshening and intrusion phases for each sampling period.

|  | FW + f4 (%) | f3 (%) | f2 (%) | f1 (%) | i1 (%) | i2 (%) | i3 (%) | SW + i4 (%) | Total (%) | N |
|---|---|---|---|---|---|---|---|---|---|---|
| July 2020 | 25 | 15 | 18 | 15 | 3 | 0 | 0 | 24 | 100 | 33 |
| April 2021 | 21 | 0 | 21 | 18 | 0 | 0 | 0 | 40 | 100 | 33 |
|  | *freshening* | | | | *intrusion* | | | | | |
|  | *proximal ← distal* | | | | *distal → proximal* | | | | | |

The distribution of the water samples from Figure 13 shows that samples from the sub-stages f1, f2 and f3 tend to move back to the seawater corner in the dry season 2021, which corresponds to a higher proportion of $Cl^-$.

## 5. Conceptual Model of the Salinity Evolution in the Luy River Catchment

In this study, 110 groundwater samples were collected in the Luy River catchment and analyzed for their hydrochemical composition. They confirm the presence of brackish to saline water in the unconsolidated coastal aquifer. The analysis of the chemical composition of the water samples brings new evidence to reveal the origin of the salinity:

1. In total, 65% of the samples exceed the WHO limit for drinking water and 100% of them have a degree of restriction on use from slight–moderate to severe in potential irrigation problems according to FAO.
2. Next to the salinity, the aquifer is locally strongly contaminated by nitrates and to a lesser extent sulfates. The origin of the nitrate is attributed to the excessive use of fertilizers in the study area. Although leaking septic tanks and sewage systems cannot be excluded, most data were collected in rural zones dominated by agricultural fields where the use of fertilizers is documented.

3.  The systematic collection of water samples in the deeper part of the aquifer has revealed the recurrent presence of saline water at depth, with a systematic increase in salinity at depth at locations where nested wells were installed. This observation is confirmed both by geophysical data acquired in the summer 2019 (Figure 14b, [34]) and interviews performed with the farmers. It confirms that previous estimation of the fresh-saline interface based only on shallow wells underestimated the actual extent.

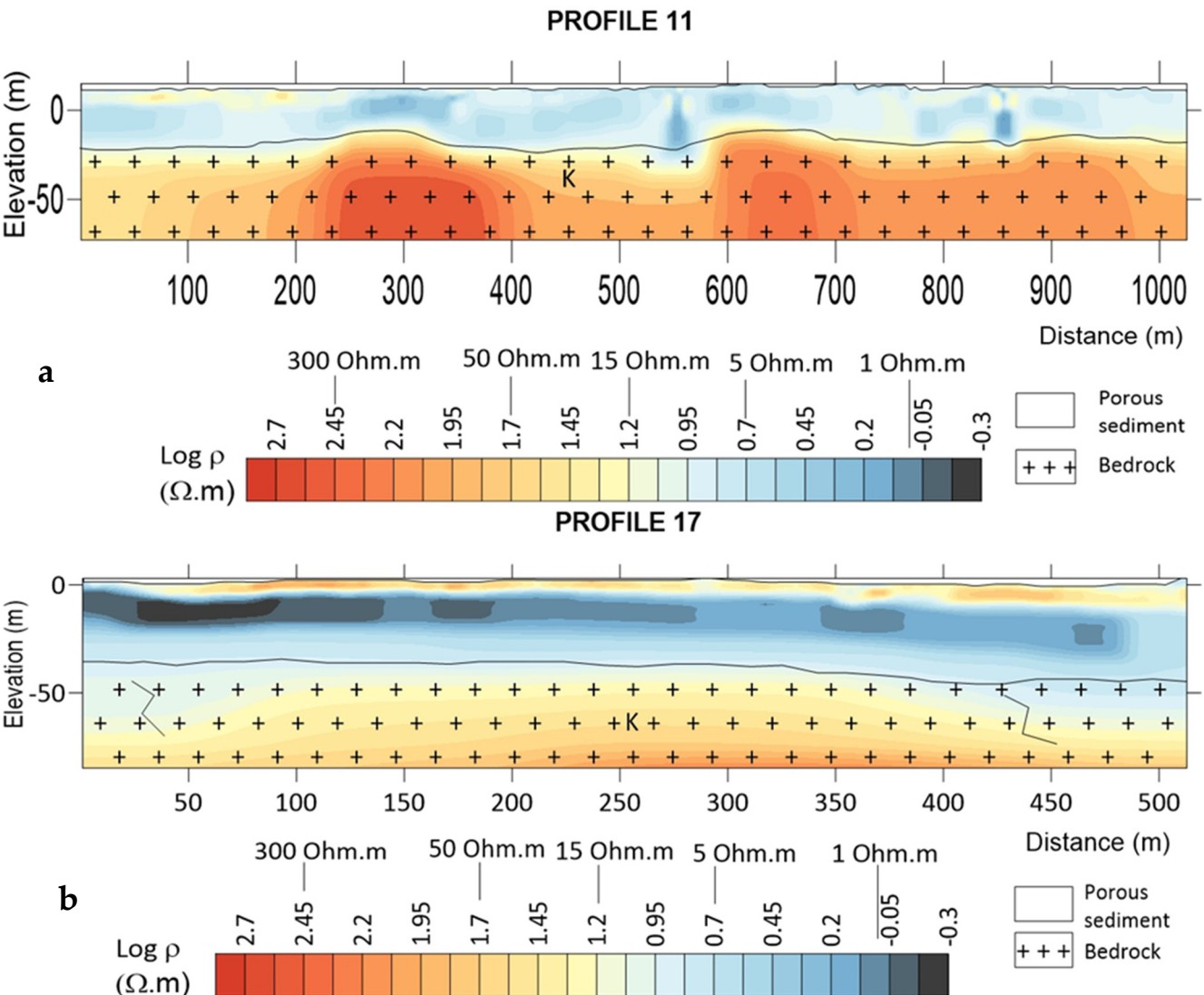

**Figure 14.** The presence of saline water at depth and strong lateral variations in salinity imaged by ERT (modified from Cong-Thi et al., 2021) [34].

4.  If the downstream and deeper part of the aquifer is systematically saline, the distribution of salinity in the upstream part is complex, with sharp variations at short distances making it difficult to predict the salinity. Saline water samples have been detected as far as 14 km inland. These observations are also confirmed by the geophysical survey (Figure 14a, [34]).

5.  The chemical composition of most of the water samples clearly shows a domination of freshening processes in the coastal aquifer. The Stuyfzand classification, the Piper diagram, the HFE diagram and the analysis of the correlation with conservative anions demonstrate that the aquifer is dominated by NaCl and $NaHCO_3$ water types with a systematic excess in $Na^+$. This points toward an enrichment of $Na^+$ related to cation exchange processes during freshening processes. This is likely the result of the natural

freshening of the aquifer since the last water high stand. The freshening of the aquifer is likely a slow process, considering that (i) the shallow aquifer is drained by the Luy River and its tributaries, (ii) the bottom of the aquifer is up to several tens of meters below sea level and (ii) the climate is semi-arid with limited recharge. Although this observation seems counterintuitive at first sight given that the local population is experiencing salinization of the water resources, we think that the presence of fossil saltwater both at depth and trapped in low permeability area can act as a secondary salinization source when the shallow and thin fresh aquifer is exploited. This is further supported by the samples located in the dune areas, which show a water composition in equilibrium. Dunes are preferential recharge zones where freshening processes can occur at a faster rate.

6. Seasonal variations in the salinity exist in the aquifer, although they are limited. Seventy-eight percent of the sampled locations experienced an increase in salinity between the summer of 2020 (wet season) and the spring of 2021 (dry season), but it was significant (TDS increases greater than 500 mg/L) only in 21% of the locations (19 samples in 90 samples recorded in 2021). This increase in salinity is characterized by a shift in the chemical composition of the sample. Although freshening processes remain dominant, the freshening signature is less pronounced. This is likely a result of the absence of recharge during the dry season, combined with the local exploitation of the shallow aquifer for agriculture and aquaculture and the presence of saltwater at depth and trapped in low permeability zones, resulting in upconing and salt migration. This is also confirmed by the experience of farmers who often see their wells becoming saltier and sometimes dry during the dry season (20 out of the 110 visited wells).

Based on the above observations, we propose a new conceptual model for the evolution of the salinity in the Luy river catchment. The simplified hydrogeological processes of relevance to saltwater intrusion and freshening are represented in Figure 15 following the CD profile direction (Figure 1).

During the last transgression, corresponding to the high water stand about 6000 years ago, the study area was largely invaded by the sea and the pores of the aquifer were filled by seawater, except in the high elevation dune area and highland (Figure 15a). When the water level started to decrease, freshwater recharge from the surface through rainfall and underground from the adjacent mountainous areas initiated the freshening of the aquifer, resulting in the current chemical signature. However, the plain is mainly made up of clays, clayey and silty sand which have a relatively low permeability. Combined with a typical semi-arid climate where recharge is limited, an important draining system through the Luy River and its tributaries, this saltwater has a long residence time and the aquifer has not reached a new equilibrium yet, especially at depth and in the low permeability zone (Figure 15b). This presence of paleo-groundwater can explain why groundwater salinity can change from fresh to brackish over short distances in clayey areas, why the deep aquifer is still saline because of density-driven flow and why salinity can increase locally when the shallow freshwater lens is exploited for irrigation. At the opposite, the dune areas have been freshened more quickly and constitute important freshwater reserves in the area.

Locally, this paleo-groundwater can act as a secondary source of salinity for saltwater intrusion when freshwater at shallow depth is extracted for irrigation—salinization occurs by upconing or lateral migration, as illustrated in Figure 15c,d. In the wet season, there is the development of a freshwater lens on top, which is likely exploited and drained by the river in the dry season, resulting in decreasing water level and upconing. Close to the Luy river, direct infiltration of saltwater is possible during high tides, especially in the dry season when the flow rate is limited. However, the large-scale spreading of river infiltration should be limited by the overall draining character of the river. Irrigation with freshwater from dams can also affect the recharge pattern and further perturb the natural situation.

This new conceptual model of the aquifer should be tested in future work. The acquisition of isotopic data should allow for the clearer identification of the freshening and salinization source as well as the age of groundwater [66,67]. Groundwater modelling could

also be used to test the hypothesis to verify if the time scale of the freshening processes is consistent with the hydraulic properties and the water balance in the catchment [68,69].

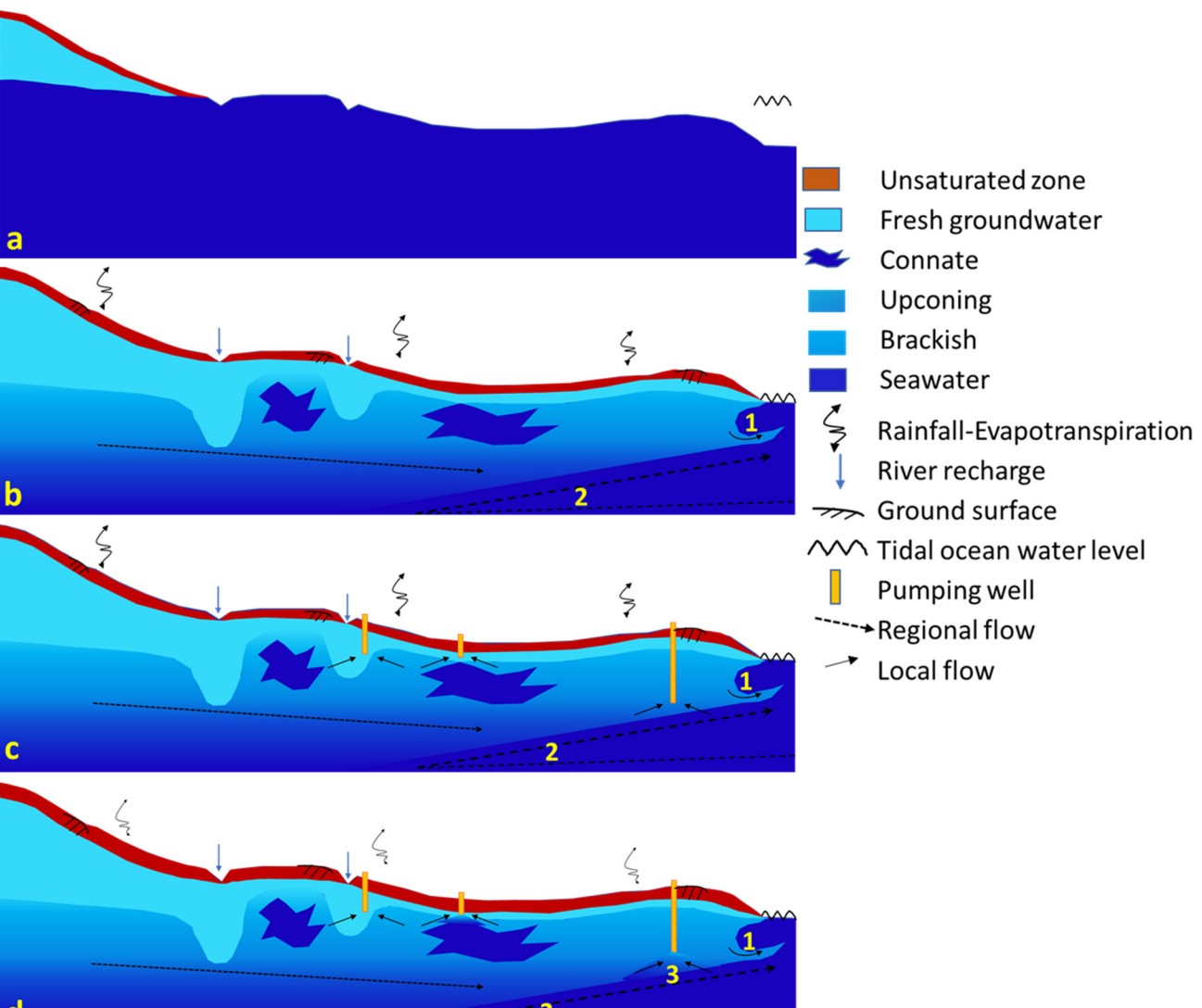

**Figure 15.** Simplified diagrams of the coastal aquifer in the Luy river catchment: 1—coastal fringe processes, such as tidal seepage face and upper seawater recirculation zone, 2—density driven circulation in the seawater zone, 3—seawater upconing due to well pumping. The saltwater intrusion and freshening processes in the different time: (**a**) Seawater covered the catchment during the past transgression, (**b**) natural freshening, (**c,d**) freshening and saltwater intrusion affected by human activities during rainy and dry seasons, respectively.

## 6. Conclusions

The hydrogeochemical analysis of 110 groundwater samples collected from the Luy river coastal aquifers in the summer of 2020 was studied and complemented by 33 new analyses made after the dry season in the spring of 2021. The findings of the study, mostly an excess in Na and K in the majority of the analyzed samples, support the current freshening trend in the aquifer. In comparison to the dry season, the freshening characteristic is more advanced in the rainy season. Salinization is stronger in the deeper aquifer and during the dry season, which is the result of a lack of recharge and exploitation of the aquifer. The new observations also confirm the observations from geophysical data revealing that (i) saline intrusion is not limited to the zone close to the river but extends further inland and at depth; (ii) salinity can vary quickly at short distance. Although the

general hydrochemical composition of groundwater was thoroughly analyzed, the isotope components of hydrogen, oxygen and carbon which can be used to estimate the origin and age of groundwater are still missing and should be investigated in the future. We also need more water samples in the deeper aquifer, as the current sampling is still biased toward shallow samples.

These analyses combined with existing data allowed us to propose a new conceptual model for the evolution of salinization in the Luy river catchment, explaining the salinity distribution, the freshening trend and the salinization during the dry season. We consider that saltwater originally in place after the last water high stand has been progressively freshened by recharge from precipitation and from the mountainous area, explaining why samples in elevated areas show freshwater facies and an equilibrium. This freshening process is not complete in the low-lying areas, at depth and in clay-rich zones where paleo-groundwater is still entrapped. This saltwater acts as a secondary salinization source during the dry season when the farmers exploit the thin shallow freshwater lens present in the Holocene aquifer.

This conceptual model is extremely important for the management of water resources in this semi-arid region of Vietnam particularly vulnerable to climate change. Combined with an analysis of the water balance (recharge: precipitation, fluxes, river and canal recharges and discharge: pumping, evaporation and river drain) and of the hydraulic conductivity of the aquifer layers, it will form the basis for the development of a groundwater model of the study area and eventually to sustainable management scenarios for this coastal region, preventing further deterioration of the groundwater resources.

In view of rising sea levels in the future due to climate change, saltwater intrusion in the river estuary will likely increase instead. It is recommended to reduce the exploitation of the aquifers and increase the surface water recharge through artificial recharge to ensure a sustainable use of the groundwater resource in this coastal region and to prevent further deterioration of quality in the shallow groundwater. The potential of the dune areas for artificial recharge and groundwater source should be investigated.

**Author Contributions:** Conceptualization, methodology, field investigation, data analysis and writing the original draft, L.P.D.; field investigation and data analysis, D.C.-T. and T.S.; project administration, writing, reviewing, editing and supervision, T.H., F.N. and H.H.H. All authors have read and agreed to the published version of the manuscript.

**Funding:** This research is funded by VLIR-UOS and the Belgian development cooperation through the grant VN2019TEA494A103, supporting the fieldwork and the PhD scholarship of the first two authors.

**Institutional Review Board Statement:** Not applicable.

**Informed Consent Statement:** Informed consent was obtained from all subjects involved in the study.

**Data Availability Statement:** The data presented in this study are available on request from the corresponding authors. The data are not publicly available yet due to the terms of the research agreement.

**Acknowledgments:** We are grateful to the VIGMR staff for their support in the field and to NAWAPI for data sharing. We would like to thank Jill Van Reybrouck for the groundwater sample analysis results and Md. Mizanur Rahman Sarker for methodology discussion.

**Conflicts of Interest:** The authors declare no conflict of interest.

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
