# Peer review of "Groundwater Salinization and Freshening Processes in the Luy River Coastal Aquifer, Vietnam"

_water, doi:10.3390/w14152358_

Round 1

Reviewer 1 Report

Comment 1: The abstract should state briefly the purpose of the research, the principal results and major conclusions. An abstract is often presented separately from the article, so it must be able to stand alone.

Comment 2: In page 3. Authors mentioned “The (1982-2019) meteorological 140data collected in the Phan Thiet station located 50 km south of the study area shows that 141 the mean monthly temperature varies from 25.2o C in January to 28.7o C in May.” Why you did not consider 2020-2022?

Comment 3: The selected area is very limited. How far can the results obtained for this area be generalized to the entire basin?

Comment 4: In the methods section, we must focus on the most important terrorist incidents that took place in the period mentioned in the study.

Comment 5: The major defect of this study is the debate or Argument is not clear stated in the introduction session. Hence, the contribution is weak in this manuscript. I would suggest the author to enhance your theoretical discussion and arrives your debate or argument.

Comment 6: Especially, the introduction section needs to re-organize. The major debate or Argument is not clear stated in the introduction session. Hence, the contribution debates are weak in this manuscript. I would suggest the author to enhance your literature discussion and arrives your debate or argument.

Comment 7: It is suggested to add articles entitled “Hazir S. Çadraku. Groundwater Quality Assessment for Irrigation: Case Study in the Blinaja River Basin, Kosovo”, “Piotr Langer. Groundwater Mining in Contemporary Urban Development for European Spa Towns”, and “El Kasri et al. Trend Analysis of Meteorological Variables: Rainfall and Temperature” to the literature review.

Comment 8: Please explain your results into steps and links to your proposed method.

Comment 9: I would like to request the author to emphasis on the contributions on practically and academically in implication session.

Comment 10: Conclusion:

•The conclusion section is currently a repeat or rehash of the preceding sections, and  needs to be re-written to improve it, keeping in mind the following suggestions.

•Update the conclusion to include the newly formulated theoretical contributions

•Mention the limitations of the study and prospects for future research.

•Summarize the key results in a compact form and re-emphasize their significance.

•Summarize how the article contributes to new knowledge in the domain.

•Provide suggestions for future studies.

•This conclusion could be worded in a manner as to emphatically motivate the academic community to get down to actionable, practical engaged scholarship.

Reviewer 2 Report

Dear authors, 

I read your paper thoroughly and found it interesting and suitable for possible publication in this journal. Please revise your paper based on:

1. please list the main novel points of this study at the end of introduction section

2. In Fig 3, mark the fresh/brackish/saline zones based on WHO. Show these maps as integrated 2D models in your area

3. In Table 3, show WHO limit for each parameter and write number of samples including % exceeding the WHO limit

4. In all figs, draw mean sea level and replace elevation with depth

5. Fig 13. Resistivity cannot be negative, please check it

6. Add more recent references related to this work 

http://dx.doi.org/10.1016/j.hydres.2020.06.001

http://dx.doi.org/10.3390/su12229730

https://doi.org/10.3390/su12093681

http://dx.doi.org/10.1111/gwat.12527

  • DOI: 
  • 10.3390/w13192719

http://dx.doi.org/10.3390/w12123408

http://dx.doi.org/10.21608/erjeng.2022.113766.1046

Reviewer 3 Report

This is an interesting manuscript providing Groundwater salinization and freshening processes in from an area of the world with particular interest. I have however some points of concern regarding the methodology followed:

1.      Authors present water analysis data collected during 2020, what is the month of sample collection? What is the impact of the rainy seasons -as authors state that seasonal distribution in the area - on the water quality? how probable sources can be identified; if possible, show the source locations details in material and methods section and also mark with different symbol of industries locations on map. Where is the geology map of the study area? What are the irrigation practices? Give in detail.

2.      Why authors not mentioned about general physicochemical parameters of the water quality (pH, EC, TH etc.? What is the impact of rainfall dissolution?

3.      In terms of sampling strategy, I would like to understand the reasoning behind stations location. Are these stations related to point and non-point pollution sources? A table with the location of each sampling station and possible impact on the water quality contamination will be helpful for the readers not familiar with the study area.

4.      In the result and discussion, no information regarding industrial and agricultural activities is given even though those seem to be major contamination sources. Show their areas and chemical concentrations in the areas and link the concentration profiles/indexes and the specific source areas as discussed in the manuscript.

5. Are they specific chemical of concerns for the Luy river? Why not include some trace? Some explanation why this methodology were selected for water quality indices will be helpful for the readers to understand the research scope.

6  I have to suggest author’s include score plots of PCA and R-mode dendrogram (cluster), the use of this plots could explain the contamination of grouped areas in these pollutants. Dendrogram plot, what about the R-mode (metals dendrogram plot) and combined with both R and Q mode to explain sources of groundwater quality.

Round 2

Reviewer 1 Report

The authors have successfully addressed all my concerns in the revised manuscript. Hence I recommend the acceptance of this paper.